# DGCBench: A Deep Graph Clustering Benchmark

**Benyu Wu**[1,2,*] **Yue Liu**[3,*] **Qiaoyu Tan**[4]**, Xinwang Liu**[3]
**Wei Du**[1]**, Jun Wang**[1]**, Guoxian Yu**[1,†]

[1]School of Software, Shandong University, Jinan, China
[2]School of Computer Science and Technology
China University of Mining and Technology, Xuzhou, China
[3]School of Computer Science and Technology
National University of Defense Technology, Changsha, China
[4]Department of Computer Science, New York University Shanghai, Shanghai, China

bywu@mail.sdu.edu.cn    yueliu@nudt.edu.cn
qiaoyu.tan@nyu.edu    xinwangliu@nudt.edu.cn
{duwei, kingjun, gxyu}@sdu.edu.cn

## Abstract

Deep graph clustering (DGC) aims to partition graph nodes into distinct clusters in an unsupervised manner. Despite rapid advancements in this field, DGC remains inherently challenging due to the absence of ground-truth, which complicates the design of effective algorithms and impedes the establishment of standardized benchmarks. The lack of unified datasets, evaluation protocols, and metrics further exacerbates these challenges, making it difficult to systematically assess and compare DGC methods. To address these limitations, we introduce DGCBench, the first comprehensive and unified benchmark for DGC methods. It evaluates 12 state-of-the-art DGC methods across 12 datasets from diverse domains and scales, spanning 6 critical dimensions: **discriminability**, **effectiveness**, **scalability**, **efficiency**, **stability**, and **robustness**. Additionally, we develop PyDGC, an open-source Python library that standardizes the DGC training and evaluation paradigm. Through systematic experiments, we reveal persistent limitations in existing methods, specifically regarding the homophily bottleneck, training instability, vulnerability to perturbations, efficiency plateau, scalability challenges, and poor discriminability, thereby offering actionable insights for future research. We hope that DGCBench, PyDGC, and our analyses will collectively accelerate the progress in the DGC community. The code is available at https://github.com/Marigoldwu/PyDGC.

## 1 Introduction

Clustering is a fundamental unsupervised learning technique that leverages the intrinsic similarity of samples to partition data into coherent groups, thereby exposing underlying structural patterns [45, 64]. Deep graph clustering (DGC) is an emerging unsupervised task that seeks to partition graph nodes into meaningful clusters, leveraging both structural and attribute information [28]. DGC has demonstrated significant potential across diverse domains, such as social network analysis [42, 56], bioinformatics [24, 50, 44], and recommendation systems [31, 32]. Harnessing the representation power of deep neural networks, DGC surpasses traditional graph clustering methods that rely solely on structure or attribute.

---

[*]Equal Contribution
[†]Corresponding author

39th Conference on Neural Information Processing Systems (NeurIPS 2025) Track on Datasets and Benchmarks.

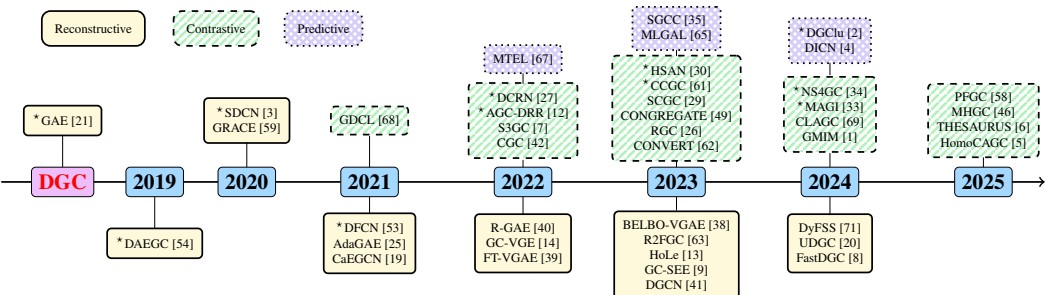

Figure 1: Evolution of DGC research: key paradigms, methods driven by graph neural networks and contrastive learning. The representative methods marked with "⋆" are benchmark methods.

However, the unsupervised nature of DGC presents intrinsic challenges. In the absence of ground-truth labels, learning discriminative and clustering-friendly node representations becomes non-trivial. Driven by the rapid progress in graph neural networks (GNNs) [22] and graph contrastive learning [68], a wide range of DGC methods have emerged [25, 38], broadly categorized into three dominant paradigms, as illustrated in Figure 1. *Reconstructive paradigm*: These methods aim to learn low-dimensional node embeddings by minimizing reconstruction loss, thus preserving the original graph structure or feature content. This paradigm emphasizes faithful representation learning, with representative methods including GAE [21], DAEGC [54], SDCN [3], DFCN [53], AdaGAE [25], CaEGCN [19], R-GAE [40], GC-VGE [14], FT-VGAE [39], BELBO-VGAE [38], R2FGC [63], HoLe [13], GC-SEE [9], DGCN [41], DyFSS [71], UDGC [20], and FastDGC [8]. *Predictive paradigm*: Instead of reconstructing input data, predictive approaches learn embeddings by forecasting inherent graph properties, such as node proximity or community affiliations. These embeddings serve as a foundation for future inference tasks. Examples include MTEL [67], SGCC [35], MLGAL [65], DGCluster [2], and DICN [4]. *Contrastive Paradigm*: Contrastive methods explicitly model semantic similarity by maximizing agreement between positive pairs and minimizing it between negatives. It focuses on learning robust and discriminative embeddings leveraging structural and semantic contrasts. Notable methods include GDCL [68], AGC-DRR [12], DCRN [27], S3GC [7], CGC [42], HSAN [30], CCGC [61], SCGC [29], CONGREGATE [49], RGC [26], CONVERT [62], MAGI [33], NS4GC [34], CLAGC [69], GMIM [1], PFGC [58], MHGC [46], THESAURUS [6], and HomoCAGC [5].

Despite this progress, the field lacks a standardized benchmark to systematically evaluate and compare DGC methods, which significantly hampers community progress. Specifically, **i**) *Fragmented datasets and inconsistent metrics*. Existing studies use datasets with varying distributions, scales, and types, and adopt inconsistent evaluation criteria. This makes results incomparable and hinders the assessment of algorithmic strengths and weaknesses. **ii**) *Insufficient and narrow evaluation*. Current evaluations overly focus on clustering accuracy while neglecting critical aspects such as **robustness**, **scalability**, **stability**, and **discriminability**. Moreover, experimental setups often lack standardized protocols, impeding reproducibility and validation.

To address these gaps and foster systematic progress in DGC, we introduce `DGCBench`, a comprehensive and unified benchmark for deep graph clustering, along with the supporting open-source toolkit `PyDGC`. Our key contributions are summarized below:

**i**) **Unified Benchmarking**. We present the first systematic DGC benchmark (`DGCBench`), which encompasses 12 diverse datasets with different characteristics and 12 state-of-the-art methods from all major paradigms. By integrating them into a standardized pipeline, we ensure fair, reproducible, and comprehensive evaluations across multiple dimensions.

**ii**) **Holistic and Multi-faceted Analysis**. Beyond conventional **effectiveness** and **efficiency** metrics, our benchmark rigorously evaluates DGC methods in terms of **robustness**, **stability**, **scalability**, and **discriminability**, revealing key weaknesses and highlighting future directions: the homophily bottleneck, training instability, vulnerability to perturbations, efficiency plateau, scalability challenges and discriminability limitations.

**iii**) **Open-source Toolkit**. We release `PyDGC`, a flexible and extensible Python library compatible with frameworks such as PyG [10] and OGB [18]. It supports the easy integration of new models and datasets, facilitating the rapid development, reproduction, and fair comparison of DGC methods.

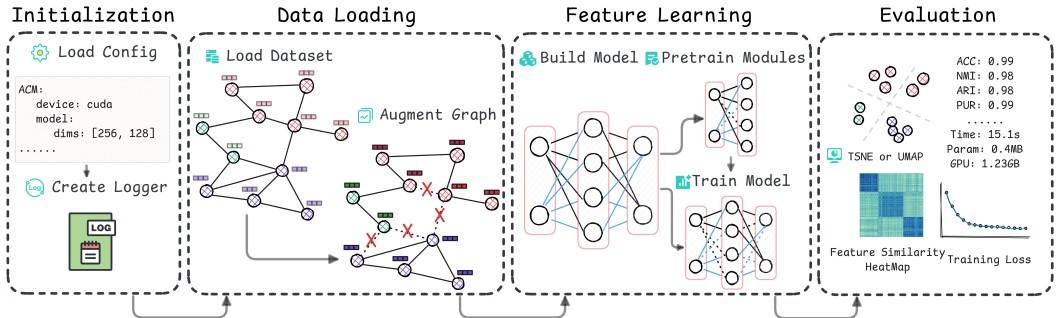

Figure 2: Our proposed standardized pipeline for DGC, includes initialization, data loading, feature learning, and evaluation, systematically covering every crucial step in the process.

## 2   Preliminary

**Notations**. Given an attributed graph $\mathcal{G}(\mathbf{A}, \mathbf{X})$, where $\mathbf{A} \in \mathbb{R}^{n \times n}$ denotes the adjacency matrix of graph $\mathcal{G}$, which characterizes the links among nodes, and $\mathbf{X} \in \mathbb{R}^{n \times d}$ represents the attribute matrix of $\mathcal{G}$, recording the $d$-dimensional attribute information of each node. In particular, for non-graph-structured data, the adjacency matrix $\mathbf{A}$ is constructed by connecting each node with its top-$k$ nearest neighbors ($k$NN). The graph $\mathcal{G}$ contains $n$ nodes and $m$ edges. The homophily of a graph can be quantified by the *homophily ratio* [70], defined as the proportion of edges connecting nodes within the same class relative to the total number of edges in the graph.

**Problem definition and unified framework**. The core objective of graph clustering is to partition the $n$ nodes in $\mathcal{G}$ into clusters based on node similarity, such that nodes within the same cluster exhibit higher mutual similarity than those in different clusters. The final label set constitutes the clustering result vector $\hat{\mathbf{y}} = [\hat{y}_1, \cdots, \hat{y}_n]^\top$. Despite the diverse paradigms in DGC, the overall process can be uniformly formalized as:

$$\hat{\mathbf{y}} = \mathcal{C}(\mathbf{Z}), \quad \mathbf{Z} = \mathcal{M}_\theta(\mathcal{G}), \quad \hat{\theta} = \arg\min_\theta \mathcal{L}(\theta), \tag{1}$$

where $\mathcal{C}$ denotes the clustering model, typically implemented as $k$-means [17] or joint clustering algorithms [57]. $\mathcal{M}_\theta$ is the feature learning module parameterized by $\theta$, responsible for extracting high-order structural and attribute dependencies from $\mathcal{G}$ to generate discriminative embeddings $\mathbf{Z}$. Distinct parameter optimization approaches give rise to different paradigms: *Reconstructive Paradigm*: $\mathcal{M}_\theta$ first learns $\mathbf{Z}$ from $\mathcal{G}$, then attempts to reconstruct the original graph data. Reconstruction can focus on node attributes [3, 55], graph topology [54, 13], or both [19, 63]. The loss function $\mathcal{L}$ combines mean-squared error for feature reconstruction and binary cross-entropy for adjacency matrix recovery. *Predictive Paradigm*: $\mathcal{M}_\theta$ predicts intrinsic properties of the graph data, including node pseudo-labels [35, 65], minimum hop counts [67], etc. By training the model to forecast these properties accurately, it learns features that reflect the underlying patterns of the graph, thereby enhancing its clustering performance. The loss function $\mathcal{L}$ incorporates a cross-entropy loss between predicted information and inherent graph properties. *Contrastive Paradigm*: It leverages contrastive learning by creating positive and negative sample pairs through graph data augmentation on $\mathcal{G}$ [7, 71]. $\mathcal{M}_\theta$ maps these samples into the feature space, where a contrastive loss function is applied, such as InfoNCE [15, 49] and Barlow Twins [66, 29], to drive positive samples close and negative samples far apart, enabling the model to learn discriminative features that benefit clustering.

**Standardized pipeline**. Building upon the unified framework described above, we establish a standardized pipeline to address the lack of standardized experimental protocols in DGC methods, as illustrated in Figure 2. This pipeline is critical for facilitating the verification of DGC methods, improving research efficiency, and elevating the overall quality of DGC studies. It provides a systematic solution that spans the entire process from initialization to final evaluation. **i)** *Initialization*. At the onset of the pipeline, a series of initialization operations is performed to ensure a unified starting state and a reliable recording mechanism for the entire process. Specifically, "Load Config" reads preset parameters and configurations. Subsequently, "Create Logger" is executed to record critical information (e.g., variations in loss values, evaluation metrics) throughout the process, which facilitates subsequent debugging and analysis. **ii)** *Data Loading*. First, "Load Dataset" imports the

Table 1: Statistics of benchmark datasets.

| Type | Dataset | # Nodes | # Edges | # Features | # Classes | Homophily |
|---|---|---|---|---|---|---|
| | Wiki | 2,405 | 17,981 | 4,973 | 17 | 0.71 |
| | Cora | 2,708 | 5,429 | 1,433 | 7 | 0.81 |
| | ACM | 3,025 | 13,128 | 1,870 | 3 | 0.82 |
| Homogeneous | Citeseer | 3,327 | 9,104 | 3,703 | 6 | 0.74 |
| | DBLP | 4,057 | 3,528 | 334 | 4 | 0.80 |
| | PubMed | 19,717 | 88,648 | 500 | 3 | 0.80 |
| | ARXIV | 169,343 | 2,315,598 | 128 | 40 | 0.65 |
| | Blog | 5,196 | 343,486 | 8,189 | 6 | 0.40 |
| Heterogeneous | Flickr | 7,575 | 479,476 | 12,047 | 9 | 0.24 |
| | Roman | 22,662 | 65,854 | 300 | 18 | 0.05 |
| $k$NN-Graph | USPS | 9,298 | 27,894 | 256 | 10 | 0.98 |
| ($k$=3) | HHAR | 10,299 | 30,897 | 561 | 6 | 0.95 |

corresponding data into the system. Subsequently, graph data augmentation ("Augment Graph") is performed as needed. For non-graph data, a $k$NN graph will be constructed. **iii**) *Feature Learning*. This constitutes one of the core stages of the pipeline, primarily tasked with learning effective representations from the graph. First, "Build Model" constructs the corresponding feature learning model according to the model type and parameters specified in the configurations. Then, various modules of the model are pretrained as required ("Pretrain Modules"). Finally, formal model training (the "Train Model") is conducted to obtain clustering results. **iv**) *Evaluation*. It conducts a comprehensive and objective evaluation of the clustering performance, including calculating clustering metrics and visualizing learned features via dimensionality reduction techniques like $t$-SNE [36] or UMAP [37], thereby enabling intuitive observation of feature distributions. Through this standardized pipeline, all aspects of DGC research can be standardized and unified, enhancing the reproducibility and comparability of studies.

## 3 DGC Benchmark

### 3.1 Benchmark Algorithms

We select 12 state-of-the-art methods as the core benchmark algorithms, covering all three paradigms. The reconstructive methods include GAE & GAE_S [21], DAEGC [54], SDCN [3], DFCN [53], the predictive method includes DGCluster [2] (abbreviated as DGClu), and the contrastive methods include AGC-DRR [12], DCRN [27], HSAN [30], CCGC [61], MAGI [33], and NS4GC [34]. For more details about benchmark algorithms, please refer to Appendix A.1. Based on the standardized pipeline, we have systematically refactored the open-source codes of these algorithms using our proposed toolkit PyDGC. We have strictly adhered to the original designs and code logic to ensure the reproducibility and comparability of the experimental results, thus providing a solid and reliable evaluation standard for DGC research.

### 3.2 Benchmark Datasets

To comprehensively evaluate the performance of algorithms, we collect 12 datasets widely used in DGC and graph self-supervised learning to construct benchmark testbeds. Our benchmark datasets have three characteristics: **i**) *comprehensive scale coverage*, encompassing small (e.g., Wiki [60], Cora, Citeseer [47], ACM [51], and DBLP [11]), medium (e.g., USPS [23] and HHAR [48]), and large (e.g., PubMed [47], Roman [43], and ARXIV [18]) graph scales to accommodate algorithm testing across multi-scale scenarios. **ii**) *diverse feature spaces*, covering low-dimensional (e.g., ARXIV and DBLP) and high-dimensional (e.g., Blog and Flickr [52]) feature spaces to enable performance validation under varying feature complexities. **iii**) *rich data type diversity*, including heterogeneous graphs (e.g., Blog, Flickr, and Roman), homogeneous graphs (e.g., Cora, ACM, and PubMed), and non-graph structured data types (e.g., USPS and HHAR), catering to diverse application scenarios in DGC. Table 1 lists the statistics of these datasets. For more details, please refer to Appendix A.2.

## 3.3 Benchmark Evaluations

We construct a holistic and systematic evaluation framework to rigorously assess benchmark algorithms across six critical dimensions:

**i) Effectiveness** is a canonical evaluation dimension in DGC, usually assessed by clustering metrics including accuracy (ACC) and normalized mutual information (NMI), which is insufficient to fully characterize algorithmic effectiveness. To address this, we expand the assessment scope. On one hand, we add assessments of the algorithm's generalization capability across graph structures with different properties (e.g., adaptability to heterogeneous graphs, high-dimensional feature graphs, and noisy graphs). On the other hand, considering the unsupervised nature of DGC, where ground-truth is often unavailable in real-world scenarios, we particularly focus on the degree to which the clustering results at algorithm termination approximate the optimal solution during the training process. Details of metrics can be found in Appendix A.3.

**ii) Robustness** mainly assesses how DGC algorithms perform under interferences like noise and missing values, which is an issue usually ignored by existing DGC studies. A robust algorithm can keep a high clustering accuracy even in a noisy setting, and its clustering outcomes will not vary greatly because of perturbations.

**iii) Stability** is evaluated through both the consistency of clustering outcomes across different random seed configurations (quantified by the standard deviation of metrics from multiple trials) and the convergence behavior of the training process (assessed via loss and NMI trajectories).

**iv) Scalability** assesses how DGC algorithms perform with large-scale graph data, focusing on their time and space complexity. A scalable algorithm can operate efficiently on extensive datasets, avoiding significant performance drops or memory overflow issues.

**v) Efficiency** evaluates the speed and resource consumption of DGC algorithms. An efficient algorithm achieves high clustering accuracy rapidly while minimizing computing resource consumption.

**vi) Discriminability** evaluates whether the feature space learned by DGC algorithms can effectively distinguish different clusters. For algorithms with high discriminability, nodes within the same cluster are more closely grouped in the feature space, and clear boundaries exist between different clusters.

## 3.4 Research Questions

**RQ1: Can the DGC algorithms effectively cluster graphs with diverse structures?** Real-world graph data exhibits significant structural diversity (e.g., heterogeneous graphs, high-dimensional features). Most current DGC algorithms rely on GNNs with homophily assumptions, limiting their generalization to non-homophilic structures. Moreover, existing studies lack validation on diverse graphs, making direct method comparisons impossible. We performed clustering experiments across 12 benchmark datasets, recording the mean and standard deviation of ACC and NMI from multiple repeated trials for both final results (algorithm-terminated outputs) and best intermediate results (optimal solutions identified during training).

**RQ2: Can the DGC algorithms obtain stable optimal clustering results?** In unsupervised scenarios, the ability of the DGC algorithm to achieve stable and optimal clustering results is critical for its practical reliability. This question directly assesses whether the algorithm can deliver consistent, high-quality partitions without ground-truth guidance, ensuring its utility in real-world applications with unknown data structures. Following the evaluation approach for stability, we recorded the training losses and NMI of 12 methods on the Cora dataset, compared the convergence of losses and NMI, and analyzed the standard deviations across different methods.

**RQ3: Are the DGC algorithms still effective when dealing with noisy data?** Real-world graph data is often disrupted by noise such as measurement errors and data missing. If an algorithm lacks noise-resistance, the reliability of its clustering results will be significantly reduced. However, most current DGC algorithms have not undergone relevant validation. Investigating the performance of DGC algorithms in noisy settings is crucial for enhancing their usability and stability in real-world scenarios. To simulate light, moderate, and severe perturbations on Cora and Blog, we introduced structural and feature-level disturbances as follows: **i)** Random edge drop/addition and feature drop were performed with probabilities of 0.2 (light), 0.5 (moderate), and 0.8 (severe); **ii)** Gaussian noise with standard deviations of 0.1 (light), 1 (moderate), and 10 (severe) was added to node features.

Then, we compared the clustering performance and the degree of result fluctuation of these algorithms on the original and the noisy data.

**RQ4: How is the comprehensive efficiency of the DGC algorithm?** The comprehensive efficiency of the DGC algorithm, including time/space complexity and resource utilization, is crucial for determining its practical application value. By comprehensively analyzing the algorithm's efficiency metrics, we can assess its suitability for real-world scenarios such as real-time analysis and high-dimensional data processing. We recorded the training time, total epochs, memory consumption, and parameter counts of 12 algorithms on datasets with different scales. Then, we analyzed the comprehensive performance to assess the efficiency.

**RQ5: Can the DGC algorithms cluster large-scale data?** As graph data scales exponentially, algorithms that cannot handle large volumes will impede technological deployment. Currently, most DGC algorithms lack evaluation on medium and large-scale data. However, evaluating the scalability of DGC algorithms in large-scale scenarios is a core requirement for promoting their application in industrial-level settings. We selected three medium-to-large-scale datasets (PubMed, Roman, and ARXIV) to compare the clustering metrics of different algorithms.

**RQ6: How discriminative is the embedding space learned by the DGC algorithms?** The discriminability of the embedding space directly impacts the quality and interpretability of clustering results. If an algorithm fails to distinguish the features of different clusters effectively, the clustering results will become meaningless. Evaluating the separability of the embedding space helps us understand the feature-learning ability of an algorithm and guides the optimization of model design. We applied $t$-SNE [36] and UMAP [37] to visualize the embeddings learned by benchmark algorithms on the Cora and ACM datasets, which helps us gain an intuitive understanding of the data distribution. $t$-SNE focuses more on preserving local structures, making it suitable for revealing fine-grained intra-cluster relationships. UMAP, while retaining local structures, also considers global topology, which is more conducive to presenting the overall distribution pattern between clusters. Moreover, we calculated discriminative indicators such as homogeneity (HOM), completeness (COM), and silhouette coefficient (SC) to quantitatively evaluate the clustering performance.

## 4 Experiments and Analyses

**Experimental Settings**. All experiments were conducted on a server equipped with a 28-core AMD CPU (116GB RAM) and an NVIDIA A10 GPU (24GB VRAM). The code was implemented using Python 3.8 and PyTorch 2.1.0. To mitigate the impact of randomness, each experiment was repeated ten times with a distinct random seed set for each run (using the current experiment iteration number as the seed value if not specified). Particularly, we only performed three repeats when the time to complete one clustering exceeded 30 minutes. As specified in [33], the maximum runtime was limited to one hour for clustering on ARXIV. Consequently, device performance can influence the quality of results. We evaluated the clustering performance of MAGI on ARXIV using a server equipped with an NVIDIA GTX 3090 (24GB VRAM) GPU and an Intel(R) Xeon(R) Gold 5318Y CPU @ 2.10GHz. For hyperparameters, if the original paper provides specific values or selection methods, they were adopted as described in the original study; otherwise, default values from the open-source code (if available) were used. Specific settings are detailed in Appendix D.

### 4.1 Effectiveness Analysis (RQ1)

**The benchmark algorithms generally suffer from a homophily bottleneck, and the training termination strategy based on fixed epochs tends to suboptimal clustering performance**. As shown in Table 2 and Table 3, different algorithms exhibit divergent performances across heterogeneous graphs, homogeneous graphs, and sparse graphs constructed by $k$NN for non-graph data: **i**) Contrastive DGC significantly outperforms other methods on heterogeneous graphs due to its strong feature discriminability, while reconstructive paradigm-based methods demonstrate greater advantages in sparse graph scenarios of non-graph data by implicitly modeling feature distributions. **ii**) Analysis of homogeneous graphs shows that, except for the ACC metric on the ACM dataset, the optimal clustering values of all methods do not significantly exceed the inherent homophily ratio of the datasets. It demonstrates that the performance of DGC algorithms closely approximates the inherent homophily ratios of datasets on most high-homophily datasets. Moreover, no significant performance improvements are observed on some datasets (e.g., DBLP, Cora, and Citeseer). This

Table 2: Clustering results on Wiki, Cora, ACM, Citeseer, DBLP, and PubMed. The "♯best" row denotes the optimal value, while other rows denote the final values at algorithm termination. The **first**, second, and *third* best performances are highlighted. "-" indicates GPU memory overflow. The darker the shade intensity of cell backgrounds, the larger the standard deviations (SD) are.

| Method | Wiki | | Cora | | ACM | | Citeseer | | DBLP | | PubMed | |
|---|---|---|---|---|---|---|---|---|---|---|---|---|
| | ACC | NMI | ACC | NMI | ACC | NMI | ACC | NMI | ACC | NMI | ACC | NMI |
| GAE | 0.4444 | 0.4485 | 0.6556 | 0.5148 | 0.8796 | 0.6205 | 0.5821 | 0.3210 | 0.4700 | 0.1907 | 0.6446 | 0.2312 |
| ♯best | 0.5301 | 0.4751 | 0.7166 | 0.5375 | 0.8803 | 0.6226 | 0.5891 | 0.3193 | 0.5101 | 0.1924 | 0.6480 | 0.2380 |
| GAE_S | 0.3913 | 0.3476 | 0.7014 | 0.5339 | 0.8507 | 0.6066 | 0.5644 | 0.3070 | 0.5165 | 0.2170 | 0.6636 | 0.2469 |
| ♯best | 0.4321 | 0.4088 | 0.7078 | 0.5431 | 0.8574 | 0.6119 | 0.5717 | 0.3091 | 0.5276 | 0.2231 | 0.6685 | 0.2597 |
| DAEGC | 0.3925 | 0.3337 | 0.6681 | 0.4854 | 0.8436 | 0.5181 | 0.6019 | 0.3345 | 0.6140 | 0.2364 | 0.6533 | 0.2432 |
| ♯best | 0.4069 | 0.3360 | 0.6860 | 0.5142 | 0.8968 | 0.6555 | 0.6143 | 0.3502 | 0.6553 | 0.2887 | 0.6533 | 0.2432 |
| SDCN | 0.1866 | 0.0303 | 0.5082 | 0.3175 | 0.8815 | 0.6373 | 0.5346 | 0.3019 | 0.6193 | 0.3176 | 0.5559 | 0.1960 |
| ♯best | 0.1934 | 0.0338 | 0.5598 | 0.3689 | 0.9004 | 0.6743 | 0.5364 | 0.3010 | 0.6349 | 0.3368 | 0.6131 | 0.2311 |
| DFCN | 0.4372 | 0.4299 | 0.4885 | 0.4058 | 0.8996 | 0.6720 | 0.5550 | 0.3630 | 0.7148 | 0.3870 | 0.4765 | 0.0734 |
| ♯best | 0.5249 | 0.4585 | 0.7020 | 0.5274 | 0.8996 | 0.6720 | 0.6836 | 0.4250 | 0.7524 | 0.4355 | 0.6742 | 0.3086 |
| DGClu | 0.5517 | 0.4574 | 0.5841 | 0.4827 | 0.7737 | 0.5217 | 0.4704 | 0.2418 | 0.3590 | 0.0586 | 0.5938 | 0.2079 |
| ♯best | 0.5843 | 0.4839 | 0.7254 | 0.5593 | 0.9042 | 0.6872 | 0.5907 | 0.3027 | 0.5450 | 0.2117 | 0.7551 | 0.3430 |
| AGC-DRR | 0.4519 | 0.4040 | 0.6681 | 0.5211 | 0.9071 | 0.7086 | 0.6579 | 0.4152 | 0.7976 | 0.4887 | 0.6232 | 0.2523 |
| ♯best | 0.4519 | 0.4040 | 0.6780 | 0.5153 | 0.9191 | 0.6835 | 0.6737 | 0.4048 | 0.8008 | 0.4942 | 0.6400 | 0.2372 |
| DCRN | 0.3472 | 0.2700 | 0.6471 | 0.5249 | 0.9183 | 0.7166 | 0.6831 | 0.4411 | 0.7053 | 0.4078 | – | – |
| ♯best | 0.4072 | 0.3261 | 0.7193 | 0.5494 | 0.9185 | 0.7162 | 0.7042 | 0.4537 | 0.7372 | 0.4339 | – | – |
| HSAN | 0.4555 | 0.4534 | 0.6390 | 0.5243 | 0.5346 | 0.2338 | 0.5347 | 0.3483 | 0.6674 | 0.3885 | – | – |
| ♯best | 0.5206 | 0.4782 | 0.7748 | 0.5912 | 0.8070 | 0.5014 | 0.6659 | 0.4150 | 0.7147 | 0.4230 | – | – |
| CCGC | 0.4682 | 0.4554 | 0.6170 | 0.4765 | 0.7062 | 0.4850 | 0.6011 | 0.3605 | 0.3382 | 0.0505 | 0.4592 | 0.0923 |
| ♯best | 0.5354 | 0.4824 | 0.7396 | 0.5640 | 0.8910 | 0.6484 | 0.6918 | 0.4332 | 0.5495 | 0.2373 | 0.6545 | 0.3167 |
| MAGI | 0.2970 | 0.2726 | 0.7324 | 0.5575 | 0.8327 | 0.5345 | 0.6819 | 0.4334 | 0.6751 | 0.3770 | 0.6181 | 0.1912 |
| ♯best | 0.5651 | 0.5046 | 0.7374 | 0.5619 | 0.9106 | 0.6972 | 0.6931 | 0.4437 | 0.7103 | 0.4075 | 0.6902 | 0.3024 |
| NS4GC | 0.4454 | 0.4263 | 0.7048 | 0.5683 | 0.7917 | 0.4871 | 0.6721 | 0.4332 | 0.7740 | 0.4544 | 0.6892 | 0.3128 |
| ♯best | 0.5237 | 0.4980 | 0.7579 | 0.5977 | 0.8882 | 0.6367 | 0.6842 | 0.4351 | 0.7919 | 0.4823 | 0.7056 | 0.3284 |
| **SD** | | 0.000 | <0.004 | <0.008 | <0.012 | <0.016 | <0.020 | <0.030 | <0.050 | <0.100 | >=0.100 | |

Table 3: Clustering results on Non-graph data, heterogeneous graphs, and ARXIV.

| Method | USPS$_{k=3}$ | | HHAR$_{k=3}$ | | Blog | | Flickr | | Roman | | ARXIV | |
|---|---|---|---|---|---|---|---|---|---|---|---|---|
| | ACC | NMI | ACC | NMI | ACC | NMI | ACC | NMI | ACC | NMI | ACC | NMI |
| GAE | 0.5808 | 0.5251 | 0.4464 | 0.4396 | 0.4658 | 0.3016 | 0.2728 | 0.1596 | 0.1612 | 0.0966 | – | – |
| ♯best | 0.6145 | 0.5246 | 0.5995 | 0.6010 | 0.5068 | 0.3308 | 0.3008 | 0.1753 | 0.1714 | 0.1009 | – | – |
| GAE_S | 0.6620 | 0.5908 | 0.4255 | 0.5017 | 0.4560 | 0.2825 | 0.2263 | 0.1256 | 0.1588 | 0.0867 | – | – |
| ♯best | 0.6620 | 0.5907 | 0.4620 | 0.5185 | 0.4702 | 0.3039 | 0.2536 | 0.1309 | 0.1696 | 0.0901 | – | – |
| DAEGC | 0.6796 | 0.6618 | 0.5800 | 0.5748 | 0.4074 | 0.2363 | 0.2643 | 0.1243 | – | – | – | – |
| ♯best | 0.6803 | 0.6597 | 0.5815 | 0.5790 | 0.4491 | 0.2737 | 0.2677 | 0.1262 | – | – | – | – |
| SDCN | 0.7320 | 0.7486 | 0.6921 | 0.7048 | 0.2404 | 0.0757 | 0.1730 | 0.0649 | 0.1461 | 0.0144 | – | – |
| ♯best | 0.7366 | 0.7441 | 0.6998 | 0.7010 | 0.3260 | 0.1329 | 0.2281 | 0.1143 | 0.1507 | 0.0128 | – | – |
| DFCN | 0.7295 | 0.7506 | 0.6718 | 0.7301 | 0.3785 | 0.2679 | 0.3113 | 0.1858 | 0.1202 | 0.0624 | – | – |
| ♯best | 0.7993 | 0.7662 | 0.7854 | 0.7731 | 0.6133 | 0.3921 | 0.3348 | 0.1904 | 0.2091 | 0.1324 | – | – |
| DGClu | 0.7128 | 0.6908 | 0.6915 | 0.7270 | 0.4506 | 0.2724 | 0.2317 | 0.1130 | 0.1154 | 0.0374 | – | – |
| ♯best | 0.7313 | 0.6786 | 0.7044 | 0.7308 | 0.4706 | 0.2868 | 0.2642 | 0.1338 | 0.1638 | 0.0579 | – | – |
| AGC-DRR | 0.6733 | 0.6791 | 0.6520 | 0.6068 | 0.7568 | 0.6548 | 0.3139 | 0.2893 | – | – | – | – |
| ♯best | 0.6900 | 0.6748 | 0.6600 | 0.6057 | 0.8369 | 0.5937 | 0.3649 | 0.2760 | – | – | – | – |
| DCRN | 0.2686 | 0.2229 | 0.3663 | 0.4696 | 0.5668 | 0.4248 | 0.1836 | 0.0894 | – | – | – | – |
| ♯best | 0.3094 | 0.2748 | 0.3888 | 0.4689 | 0.8363 | 0.6683 | 0.2887 | 0.2055 | – | – | – | – |
| HSAN | – | – | – | – | 0.4540 | 0.3098 | – | – | – | – | – | – |
| ♯best | – | – | – | – | 0.4900 | 0.3202 | – | – | – | – | – | – |
| CCGC | 0.3882 | 0.3620 | 0.3938 | 0.4193 | 0.2942 | 0.1095 | 0.1825 | 0.0753 | 0.1626 | 0.0932 | – | – |
| ♯best | 0.5556 | 0.4852 | 0.5389 | 0.5470 | 0.3551 | 0.1516 | 0.1972 | 0.0051 | 0.1874 | 0.1135 | – | – |
| MAGI | 0.5703 | 0.5398 | 0.4504 | 0.4593 | 0.4172 | 0.2444 | 0.3121 | 0.1606 | 0.1790 | 0.0694 | 0.3903 | 0.4684 |
| ♯best | 0.6650 | 0.5864 | 0.6927 | 0.6556 | 0.5129 | 0.3039 | 0.3446 | 0.1889 | 0.1966 | 0.1220 | – | – |
| NS4GC | 0.6957 | 0.6688 | 0.6056 | 0.5588 | 0.3636 | 0.2077 | 0.3151 | 0.0184 | 0.1442 | 0.0808 | – | – |
| ♯best | 0.7346 | 0.6719 | 0.6367 | 0.5780 | 0.4398 | 0.2612 | 0.3729 | 0.2198 | 0.1724 | 0.1152 | – | – |
| **SD** | | 0.000 | <0.004 | <0.008 | <0.012 | <0.016 | <0.020 | <0.030 | <0.050 | <0.100 | >=0.100 | |

phenomenon suggests that intrinsic homophily may constitute a potential upper bound for clustering performance, where clustering results can hardly surpass the level determined by the homophilic structure of data. **iii**) The widespread discrepancies between the final clustering results and best training results of each algorithm indicate that the fixed-epoch termination strategy fails to ensure the model converges to a stable optimal solution, highlighting the necessity of introducing adaptive termination mechanisms to improve clustering stability. **iv**) Figure 3 illustrates the loss trajectories of 12 methods during training, *revealing a counterintuitive phenomenon: SDCN, DFCN, and HSAN exhibit rising loss values throughout training, yet still achieve competitive clustering performance*. This observation contradicts the fundamental assumption that minimizing the loss aligns with improving clustering quality, suggesting potential flaws in the traditional loss-clustering performance correlation, or unrecognized mechanisms enabling effective feature learning despite increasing loss. For all clustering results, please refer to Appendix B.1.

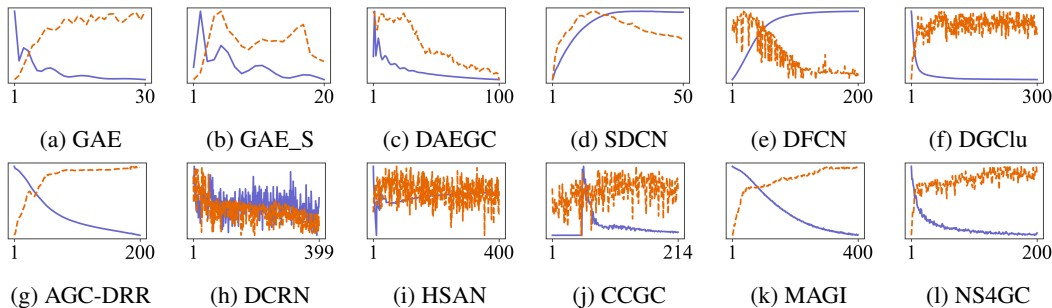

Figure 3: Loss and NMI curves of 12 benchmark algorithms on the Cora.

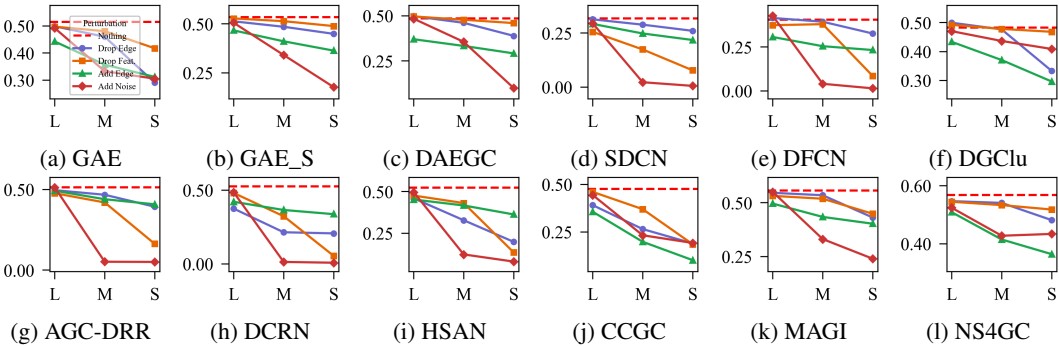

Figure 4: NMI of 12 methods on Cora with light (L), moderate (M), and severe (S) perturbations.

## 4.2 Stability Analysis (RQ2)

**Existing algorithms generally face stability challenges characterized by "attainable upper bounds but uncontrollable convergence paths"**. Visual inspection of standard deviations via the cell backgrounds in Table 2 and Table 3 reveals that the standard deviations of optimal clustering metrics (♯best) for most methods are significantly smaller than those of their final convergence results. This phenomenon indicates that while these algorithms can reach relatively stable performance ceilings during training (i.e., optimal values exhibit high stability), their convergence processes lack robustness, making it difficult to repeatedly achieve theoretical optimal solutions through fixed-epoch training strategies. Further analysis of loss and NMI curves in Figure 3 shows that most methods demonstrate stable training trends with monotonic or oscillatory convergence. However, clustering accuracy does not converge stably in tandem with loss minimization, *which suggests a potential misalignment between clustering objectives and feature learning goals*.

## 4.3 Robustness Analysis (RQ3)

**While all DGC algorithms degrade under moderate-to-severe noise, some unexpectedly demonstrate resilience or marginal performance improvements under light perturbations**. Figure 4 illustrates the trends of the clustering metric NMI for 12 methods on the CORA dataset as the perturbation intensity increases. Notably, all methods exhibit significant performance degradation with rising noise levels, though the sensitivity varies by noise type: **i**) Adding Gaussian noise induces drastic declines in NMI when transitioning from light to moderate perturbation levels. **ii**) Dropping feature leads to sharp drops in performance during the moderate-to-severe noise regime. *A surprising observation is that under light perturbations, some methods show no obvious degradation and even slight improvements*. For example, **i**) DAEGC demonstrates marginal performance gains when randomly dropping features. **ii**) DFCN exhibits slight NMI increases under little Gaussian noise. **iii**) DGClu shows improved clustering results under light edge-dropping perturbations. Additional robustness analysis on a heterogeneous graph Blog can be found in Appendix B.2.

Table 4: Efficiency metrics on Cora, Citeseer, and PubMed. ↑ indicates that larger values are preferable, while ↓ indicates the opposite.

| | | GAE | GAE_S | DAEGC | SDCN | DFCN | DGClu | AGC-DRR | DCRN | HSAN | CCGC | MAGI | NS4GC |
|---|---|---|---|---|---|---|---|---|---|---|---|---|---|
| **NMI** ↑ | Cora | 0.51 | 0.53 | 0.49 | 0.32 | 0.41 | 0.48 | 0.52 | 0.52 | 0.52 | 0.48 | 0.56 | 0.57 |
| | Citeseer | 0.32 | 0.31 | 0.33 | 0.30 | 0.36 | 0.24 | 0.40 | 0.44 | 0.35 | 0.36 | 0.43 | 0.43 |
| | PubMed | 0.23 | 0.25 | 0.24 | 0.20 | - | 0.21 | - | - | - | 0.09 | 0.19 | 0.31 |
| **Speed** (its/s) ↑ | Cora | 56.44 | 36.52 | 43.87 | 32.60 | 19.73 | 89.30 | 2.60 | 19.73 | 12.29 | 33.55 | 80.12 | 64.81 |
| | Citeseer | 34.46 | 24.93 | 28.89 | 23.08 | 15.96 | 77.47 | 1.77 | 15.96 | 8.63 | 28.13 | 44.31 | 56.00 |
| | PubMed | 1.55 | 1.07 | 1.00 | 4.88 | - | 17.98 | 0.11 | - | - | 14.22 | 2.75 | 4.21 |
| **Time** (s) ↓ | Cora | 0.53 | 0.55 | 2.28 | 1.53 | 20.27 | 3.36 | 76.91 | 20.27 | 32.55 | 11.92 | 4.99 | 3.09 |
| | Citeseer | 0.87 | 0.80 | 3.46 | 2.17 | 25.06 | 3.87 | 113.31 | 25.06 | 46.33 | 14.22 | 9.03 | 0.89 |
| | PubMed | 19.37 | 18.76 | 100.29 | 10.26 | - | 16.68 | 1802.37 | - | - | 62.10 | 145.47 | 47.46 |
| **GPU** (MB) ↓ | Cora | 169.10 | 172.39 | 339.25 | 735.32 | 297.11 | 157.62 | 494.30 | 962.31 | 3526.05 | 2976.50 | 257.21 | 255.71 |
| | Citeseer | 248.16 | 250.32 | 529.43 | 817.16 | 422.78 | 179.94 | 669.59 | 1364.12 | 13226.41 | 2976.50 | 372.74 | 450.04 |
| | PubMed | 6023.24 | 6027.89 | 15702.99 | 6398.37 | - | 3058.03 | 17462.94 | - | - | 10140.18 | 8038.78 | 9591.20 |
| **Param** (M) ↓ | Cora | 0.37 | 0.37 | 0.37 | 5.97 | 0.48 | 0.41 | 1.60 | 0.48 | 9.63 | 1.43 | 0.73 | 0.38 |
| | Citeseer | 0.95 | 0.95 | 0.95 | 9.38 | 0.48 | 0.99 | 1.60 | 0.48 | 11.49 | 3.70 | 2.16 | 0.95 |
| | PubMed | 0.13 | 0.13 | 0.13 | 4.57 | 0.48 | 0.17 | 1.60 | 0.48 | 20.22 | 0.50 | 0.19 | 0.19 |
| **Avg.** ↓ | | 3.53 | 3.93 | 5.80 | 7.40 | 7.73 | 4.13 | 9.20 | 7.73 | 10.33 | 7.87 | 5.27 | 3.67 |

Table 5: Discriminability metrics of 12 benchmark algorithms on the Cora.

| | GAE | GAE_S | DAEGC | SDCN | DFCN | DGClu | AGC-DRR | DCRN | HSAN | CCGC | MAGI | NS4GC |
|---|---|---|---|---|---|---|---|---|---|---|---|---|
| **HOM** ↑ | 0.542 | 0.547 | 0.521 | 0.358 | 0.533 | 0.551 | 0.530 | 0.547 | 0.593 | 0.563 | 0.567 | 0.575 |
| **COM** ↑ | 0.533 | 0.539 | 0.507 | 0.382 | 0.523 | 0.568 | 0.513 | 0.578 | 0.590 | 0.565 | 0.557 | 0.562 |
| **SC** ↑ | 0.281 | 0.347 | 0.330 | 0.213 | 0.199 | 0.424 | 0.526 | 0.152 | 0.110 | 0.089 | 0.189 | 0.276 |
| **Avg.** ↓ | 7.0 | 5.7 | 8.7 | 10.3 | 9.0 | 3.0 | 7.0 | 6.7 | 4.3 | 6.3 | 6.0 | 4.0 |

### 4.4 Efficiency Analysis (RQ4)

**The comprehensive efficiency of most DGC algorithms is even lower than that of the vanilla GAE.** Multidimensional comparisons in Table 4 demonstrate that the classical method GAE achieves average optimal efficiency with lightweight model parameters, fast runtime, and stable clustering performance. This reveals that *contemporary DGC algorithms overlook efficiency optimization when pursuing clustering accuracy*. Balancing clustering precision with computational efficiency and designing lightweight models for large-scale graphs remain critical challenges for future research.

### 4.5 Scalability Analysis (RQ5)

**Current graph clustering methods widely face scalability limitations**. As shown in Table 2 and Table 3, algorithms marked with "-" experienced GPU memory overflow on a 24GB GPU, particularly evident for the medium-to-large challenging datasets, Roman and ARXIV, and the HSAN algorithm. Specifically, HSAN's requirement for $\mathcal{O}(n^2)$ storage due to feature learning on adjacency matrices leads to drastic GPU memory consumption increases. While MAGI handles large-scale data through sampling strategies [16], such approaches suffer from insufficient global graph structure learning, failing to meet the accuracy demands of reconstructive methods for structural modeling.

### 4.6 Discriminability Analysis (RQ6)

**DGC algorithms demonstrate excellent cohesion within clusters in embeddings but exhibit notably insufficient inter-cluster separability, particularly in distinguishing samples at the cluster boundaries**. As shown in Figure 5, the embeddings learned by 12 methods on the Cora reveal that while most intra-cluster samples form compact clusters, fuzzy inter-cluster boundaries and numerous cross-cluster mixing points are evident. Moreover, quantitative metrics in Table 5 further confirm that despite structural variations across methods, the embeddings generally exhibit strong cohesion but weak separability. This discrepancy highlights the fundamental limitations of DGC models in capturing complex structures with clear inter-cluster decision boundaries. Additional visualization on the ACM dataset can be found in Appendix B.3.

## 5 Conclusions and Future Directions

We introduce DGCBench, a comprehensive and unified benchmarking framework for deep graph clustering, which holistically evaluates 12 state-of-the-art methods across three paradigms from six dimensions on 12 datasets with diverse characteristics. We provide a standardized experimental

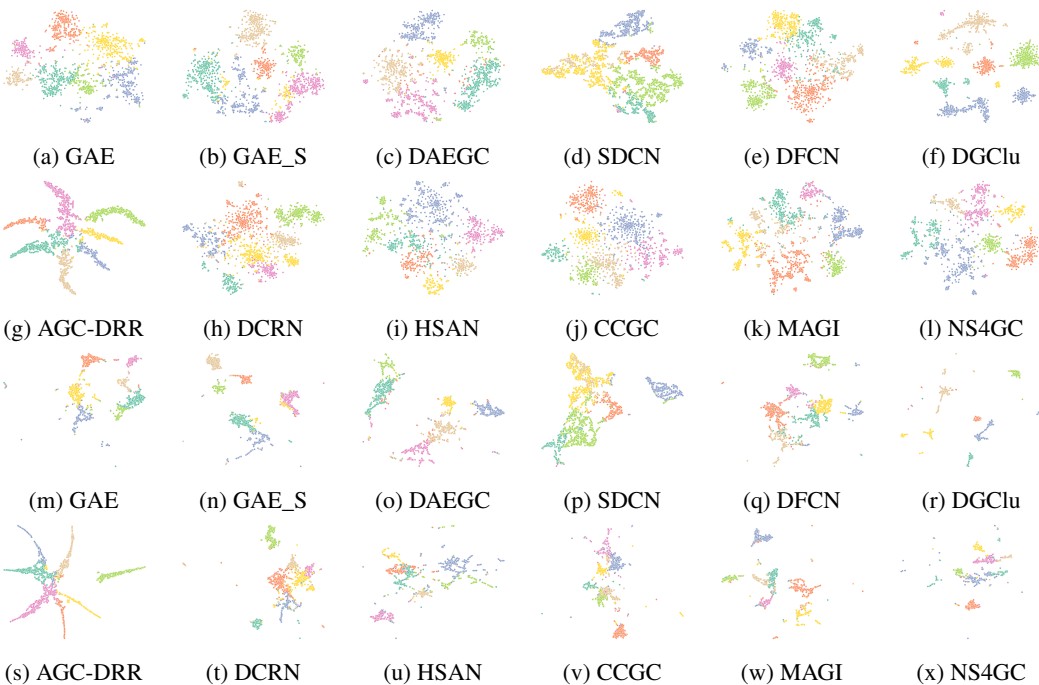

Figure 5: $t$-SNE (first and second rows) and UMAP (third and fourth rows) 2D visualization of embeddings learned by 12 benchmark algorithms on Cora.

pipeline and an open-source tool, `PyDGC`. Extensive experimental results and analyses uncover critical research directions for DGC:

**i) Homophily Bottleneck**: Although the homophily assumption enabled early GNN-based clustering methods to achieve better results than those that did not consider such relationships, this assumption now limits the upper bound of clustering performance. Therefore, there is an urgent need to explore methods that weaken the reliance on the homophily assumption.

**ii) Stability Deficiency**: It is manifested in sensitivity to initial parameters and unstable convergence during the training process, which reduces the reliability of the algorithm and restricts its practical application. It is urgent to explore convergence methods with strict theoretical support.

**iii) Robustness Gap**: The message passing mechanism of GNNs can accelerate the diffusion of noise, making the model sensitive to severe noise. Future research should be conducted in directions such as noise source suppression, the design of anti-noise aggregation methods, and result calibration.

**iv) Efficiency Plateau**: Despite years of research, DGC methods have not achieved substantial progress in comprehensive efficiency. In the future, DGC algorithms that balance accuracy and efficiency should be studied to ensure that the research outcomes can be applied to practical applications.

**v) Scalability Challenges**: DGC faces challenges in scaling on large-scale datasets, with high computational and storage costs, hindering its deployment in resource-constrained environments. Although sampling enables large-scale graph learning, balancing global structures in sampling and implementing clustering still needs to be explored.

**vi) Discriminability Limitations**: Although the learned embeddings exhibit strong intra-cluster cohesion, their inter-cluster discriminability is significantly poor. This indicates that future efforts should focus on the feature learning of clustering boundaries and hard samples.

These insights aim to inspire the DGC research community and foster technological advancements in the field. Moving forward, we plan to expand `DGCBench` and `PyDGC` by incorporating real-world datasets (e.g., single-cell sequencing, hyperspectral images) and updating them with the latest models. We also sincerely welcome suggestions from researchers to collectively advance DGC methodologies.

## Acknowledgments

This work was supported by the National Key Research and Development Program of China (No. 2024YFF1206604), NSFC (62432006 and 62272276), Shandong Provincial Natural Science Foundation (No. ZR2024JQ001), Taishan Scholars Program (No. tsqn202306007 and tsqn202408317), Postdoctoral Innovation Program of Shandong Province (No. SDCX-ZG-202501019), China Postdoctoral Science Foundation (No. 2025M771502).

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

# A  Details of DGC Benchmark

## A.1  Details of Benchmark Algorithms

**GAE & GAE-SSC** (abbreviated as GAE_S) [21]: GAE belongs to the reconstructive paradigm. It encodes the structural information of graph data by stacking graph convolutional networks (GCN) as layers, and learns the features of the graph by reconstructing the adjacency relationships between nodes. The model is trained by minimizing the discrepancy between the reconstructed graph and the input graph. Finally, $k$-means is used to cluster the embeddings. GAE_S is a variant of GAE, which treats clustering as one of the optimization objectives and incorporates self-supervised clustering loss [57], thereby infusing clustering guidance into feature learning.

**DAEGC** [54]: DEAGC is categorized within the reconstructive paradigm and shares the overall architecture with GAE-SSC. Where it differs is in its adoption of graph attention networks as the backbone network, thereby placing greater emphasis on node attribute information.

**SDCN** [3]: SDCN falls under the reconstructive paradigm. Unlike the aforementioned methods, it encodes attribute information by introducing an autoencoder. The model leverages GCN to fuse attribute and structural information, thereby enhancing the expressive power of embedding representations. Additionally, SDCN incorporates a self-supervised clustering loss for joint optimization.

**DFCN** [53]: DFCN falls within the reconstructive paradigm. Building upon SDCN, the model introduces a dynamic fusion mechanism to adaptively fuse attribute and structural information. Specifically, DFCN leverages a dynamic cross-modal fusion mechanism and a triple self-supervised strategy to efficiently utilize graph structural information and node attribute information.

**DGCLUSTER** (abbreviated as DGClu) [2]: DGClu adheres to the predictive paradigm for graph clustering. First, the model employs GNNs to extract node embeddings. Second, it leverages node attributes or additional information to evaluate the similarity between embedding vectors, thereby guiding the feature learning process. Finally, the clustering result is obtained by conducting Birch clustering on the embeddings.

**DCRN** [27]: DCRN is categorized within the contrastive paradigm, specifically addressing the representation collapse issue that arises during the node encoding process in existing methods. Specifically, the model employs a siamese network architecture to encode samples from different views, and performs identity matrix approximation on both the cross-view sample correlation matrix and the cross-view feature correlation matrix. This approach reduces information redundancy from both sample and feature dimensions, thereby significantly enhancing the discriminative power of features. Additionally, DCRN introduces a propagation regularization term, enabling shallow networks to capture long-range dependency information.

**AGC-DRR** [12]: AGC-DRR falls under the contrastive paradigm. It adaptively learns a redundant edge dropout matrix by introducing a contrastive learning mechanism, ensuring the diversity of compared samples to reduce redundancy in the input space. Meanwhile, it enforces the correlation matrix of cross-augmented sample embeddings to approximate an identity matrix, thereby minimizing redundant information in the latent space.

**HSAN** [30]: HSAN falls under the contrastive paradigm. It calculates the similarity between samples based on attribute embeddings and structural embeddings, respectively, assisting in computing the hardness degree of samples, and leverages high-confidence clustering information to guide the design of weight functions. After identifying positive and negative samples, the model dynamically increases the weights of hard samples and reduces the weights of easy samples, thereby effectively enhancing its discriminability for samples.

**CCGC** [61]: CCGC falls under the contrastive paradigm. To address the issue of negative sample selection lacking clustering information guidance, it selects positive samples from the same high-confidence cluster across two views and uses the centers of different high-confidence clusters as negative samples, thereby enhancing the discriminability and reliability of the constructed sample pairs.

**MAGI** [33]: MAGI is categorized within the contrastive paradigm, leveraging modularity maximization as a contrastive pretext task to effectively uncover the latent information of communities while mitigating semantic drift. Furthermore, MAGI adopts mini-batch training, enabling remarkable scalability.

**NS4GC** [34]: NS4GC falls under the contrastive paradigm. It performs view augmentation by randomly dropping edges and masking features, and leverages a shared-parameter GNN encoder to extract node representations. The model constructs a latent node similarity matrix via cross-view cosine similarity, ensures the accuracy of node similarity through node neighbor alignment, and achieves effective sparsity using semantic-aware sparsification, thus making node similarity both accurate and effectively sparse.

## A.2  Details of Benchmark Datasets

**Wiki**[60]: The Wiki is a Web hyperlink network, and its topological structure can effectively simulate the complex associations between web pages. It contains 2,405 documents and 17,981 links, encompassing 17 categories. Features are constructed from the TF-IDF of documents, comprising 4,973 features.

**Cora** [47]: The Cora dataset contains 2,708 scientific publications classified into seven classes, with a citation network of 5,429 links. Each publication is represented by a 1,433-dimensional binary word vector indicating the presence or absence of 1,433 unique dictionary words.

**ACM** [51]: The ACM dataset contains a paper network comprising 3,025 papers and 13,128 edges, where edges represent co-authorship. Paper features are 1,870-dimensional bag-of-words vectors of keywords. Based on research areas, papers are divided into three classes: database, wireless communication, and data mining.

**Citeseer** [47]: The Citeseer dataset contains 3,327 scientific publications classified into six classes, with a citation network of 9,104 links. Each publication is represented by a 3,703-dimensional binary word vector indicating the presence or absence of 3,703 unique dictionary words.

**DBLP** [11]: The DBLP dataset contains an author network of 4,057 authors with 3,528 edges, where edges represent co-authorship and two authors are connected if they have collaborated. Authors are classified into four research areas, including database, data mining, machine learning, and information retrieval, with labels assigned based on their published conferences. Author features are represented as 334-dimensional bag-of-words vectors constructed from keywords.

**PubMed** [47]: The PubMed dataset contains 19,717 scientific publications on diabetes from the PubMed database, classified into three classes, with a citation network of 88,648 links. Each publication is represented by 500-dimensional TF/IDF weighted word vectors, where each vector indicates term presence weighted by inverse document frequency across a 500-word dictionary.

**Ogbn-arxiv** [18]: The ogbn-arxiv dataset is a directed graph representing the citation network of 169,343 Computer Science arXiv papers indexed by MAG, with 2,315,598 directed edges indicating citation relationships. Each paper is characterized by a 128-dimensional feature vector derived from averaging word embeddings of its title and abstract, where embeddings are trained via the skip-gram model on the MAG corpus. The prediction task involves classifying papers into 40 manually labeled arXiv CS subject areas (e.g., cs.AI, cs.LG), formulated as a 40-class classification problem to automate scientific publication topic categorization.

**BlogCatalog** (abbreviated as Blog) [52]: The Blog contains 5,196 nodes, 343,486 edges, and covers 6 interest categories, which are collected from BlogCatalog. The features matrix has 8,189 columns.

**Flickr** [52]: Collected from its namesake online community platform, the Flickr dataset includes 9 interest groups as label categories. The relationship network of 7,575 subscribed users contains 479,476 edges, with user features represented as 12,047 dimensions.

**Roman-empire** (abbreviated as Roman) [43]: The Roman dataset is constructed from the English Wikipedia article "Roman Empire" (one of the longest Wikipedia articles), featuring 22,662 nodes, 65,854 edges, 300-dimensional fastText word embeddings as node features, and 18 labeled syntactic roles. It is designed to test GNNs in scenarios with low homophily and sparse connectivity for handling semantic dependencies.

**USPS** [23]: The USPS dataset consists of 9,298 grayscale handwritten digit images (0-9), each with a 16×16 pixel resolution. The dataset originates from real-world handwritten digits on envelopes, such as postal codes and phone numbers, making it a classic benchmark for optical character recognition and machine learning tasks.

**HHAR** [48]: The HHAR dataset comprises 10,299 sensor measurements from smartphones and smartwatches, partitioned into six human activity categories: biking, sitting, standing, walking, stair up, and stair down.

## A.3 Metrics

**Clustering Accuracy (ACC)**: The ACC is used to measure the correspondence between clustering labels and ground-truth labels. Its calculation is as follows:

$$\text{ACC} = \sum_{i=1}^{n} \frac{\Phi\left(\mathbf{y}_i, \text{map}(\hat{\mathbf{y}}_i)\right)}{n}, \tag{2}$$

where $\mathbf{y}_i$ is the ground-truth label of the node $i$ and map$(\hat{\mathbf{y}}_i)$ represents the clustering label after Hungarian label matching. $\Phi$ is a counting function that increments the count by 1 if the labels are identical.

**Normalized Mutual Information (NMI)**: The NMI is used to measure the degree of information sharing between the distributions of clustering labels and ground-truth labels, reflecting the amount of ground-truth information inferable from clustering labels. Ranging from 0 to 1, where 0 indicates complete irrelevance and 1 signifies perfect consistency, it is frequently employed in scenarios where the number of clustering categories differs from that of ground-truth categories. The calculation is as follows:

$$\text{NMI} = \frac{2 \cdot \text{MI}(\mathbf{y}, \hat{\mathbf{y}})}{H(\mathbf{y}) + H(\hat{\mathbf{y}})}, \ \text{MI} = \sum_{i=1}^{n} \sum_{j=1}^{n} P(i,j) \log \frac{P(i,j)}{P(i)P'(j)}, \tag{3}$$

where $H(\cdot)$ is entropy, which is the amount of uncertainty for a partition set. $P(i)$ is the probability that an object picked at random from $\mathbf{y}$ falls into class $\mathbf{y}_i$. $P'(j)$ is the probability that an object picked at random from $\hat{\mathbf{y}}$ falls into class $\hat{\mathbf{y}}_j$. $P(i,j)$ is the probability that an object picked at random falls into both classes $\mathbf{y}_i$ and $\hat{\mathbf{y}}_j$.

**Adjusted Rand Index (ARI)**: The ARI measures the agreement between clustering results and ground-truth labels, with values ranging from [-1, 1]: 1 indicates a perfect match, 0 indicates performance equivalent to random assignment, and -1 indicates worse-than-random clustering. ARI is particularly suitable for imbalanced class distributions, and is given by:

$$\text{ARI} = \frac{\text{RI} - E[\text{RI}]}{\max(\text{RI}) - E[\text{RI}]}, \ \text{RI} = \frac{a+b}{\frac{1}{2}n(n-1)}, \tag{4}$$

where $E[\text{RI}]$ denotes the mathematical expectation of the Rand Index. $a$ represents the number of node pairs that are assigned to the same cluster in both $\mathbf{y}$ and $\hat{\mathbf{y}}$, while $b$ represents the number of node pairs that are assigned to different clusters in both $\mathbf{y}$ and $\hat{\mathbf{y}}$.

**Macro F-score (F1)**: The F1 is the harmonic mean of precision and recall calculated between clustering labels and ground-truth labels, used to comprehensively evaluate clustering performance. The F1 is given by:

$$\text{F1} = \frac{2 \times \text{precision} \times \text{recall}}{\text{precision} + \text{recall}}, \ \text{precision} = \frac{\text{TP}}{\text{TP} + \text{FP}}, \ \text{recall} = \frac{\text{TP}}{\text{TP} + \text{FN}}, \tag{5}$$

where TP denotes true positive samples, FP denotes false positive samples, FN denotes false negative samples.

**Homogeneity (HOM)**: The desirable objective of HOM is that each cluster contains only members of a single class. The HOM is formally given by:

$$\text{HOM} = 1 - \frac{H(C|K)}{H(C)}, \tag{6}$$

where $H(C|K)$ is the conditional entropy of the classes given the cluster assignments. $H(C)$ is the entropy of the classes.

**Completeness (COM)**: The desirable objective of COM is that all members of a given class are assigned to the same cluster. The COM is formally given by:

$$\text{COM} = 1 - \frac{H(K|C)}{H(K)}, \tag{7}$$

where $H(K|C)$ is the conditional entropy of the cluster assignments given the classes. $H(K)$ is the entropy of the clusters.

**Silhouette Coefficient (SC)**: A higher SC indicates a model with more distinct clusters. The SC is computed for each sample and consists of two components: **i)** $a$ denotes the mean distance between a sample and all other points in its same cluster; **ii)** $b$ denotes the mean distance between a sample and all other points in the nearest neighboring cluster. The SC is then given as:

$$\text{SC} = \frac{1}{n} \sum_{i=1}^{n} \frac{b_i - a_i}{\max(a_i, b_i)}. \tag{8}$$

# B  Additional Experimental Results

## B.1  Additional Results of Other Metrics

Building on the effectiveness analysis of the DGC methods across different dataset types using ACC and NMI in Section 4.1, this section further introduces core clustering evaluation metrics ARI and F1-score, with experimental results presented in Tables 6 and 7. These metrics further highlight the homophily bottleneck in existing DGC methods. Section 4.6 reveals a typical characteristic of current DGC methods through a dual validation system combining visualization and quantitative metrics: while they can learn highly cohesive sample representations, these representations have significant discriminability limitations. Tables 8 to 10 comprehensively display three key metrics (i.e., COM, HOM, and SC) for 12 comparative methods across 12 benchmark datasets. Joint analysis of HOM and COM shows that contrastive learning-based methods have notable advantages. Combined with the SC metric, AGC-DRR performs better on homogeneous graphs, DGClu achieves better results on heterogeneous graphs, while DFCN is more suitable for clustering non-graph.

## B.2  Additional Robustness Results on Blog

In Section 4.3, we analyze the performance of 12 methods under four perturbation types and three perturbation intensities on the CORA dataset with a high homophily ratio. Experiments show that existing clustering methods

Table 6: The ARI of 12 benchmark methods on 12 datasets.

| Method | Wiki | Cora | ACM | Citeseer | DBLP | PubMed | ARXIV | USPS | HHAR | Blog | Flickr | Roman |
|---|---|---|---|---|---|---|---|---|---|---|---|---|
| GAE | 0.2974 | 0.4480 | 0.6784 | 0.3223 | 0.1694 | 0.2258 | – | 0.4453 | 0.3172 | 0.2386 | 0.0973 | 0.0476 |
| ♯best | 0.3459 | 0.4942 | 0.6802 | 0.3224 | 0.1764 | 0.2341 | – | 0.4548 | 0.4742 | 0.2691 | 0.0984 | 0.0545 |
| GAE_S | 0.2352 | 0.4758 | 0.6535 | 0.2944 | 0.1976 | 0.2518 | – | 0.5177 | 0.3111 | 0.2183 | 0.0476 | 0.0415 |
| ♯best | 0.2861 | 0.4914 | 0.6595 | 0.2991 | 0.2090 | 0.2611 | – | 0.5177 | 0.3230 | 0.2468 | 0.0895 | 0.0486 |
| DAEGC | 0.1695 | 0.4200 | 0.5861 | 0.3269 | 0.2124 | 0.2408 | – | 0.5815 | 0.4795 | 0.1653 | 0.0781 | – |
| ♯best | 0.1769 | 0.4496 | 0.7189 | 0.3567 | 0.2356 | 0.2408 | – | 0.5790 | 0.4830 | 0.2078 | 0.0824 | – |
| SDCN | 0.0116 | 0.2640 | 0.6869 | 0.2757 | 0.3250 | 0.1615 | – | 0.6629 | 0.5934 | 0.0271 | 0.0240 | 0.0046 |
| ♯best | 0.0192 | 0.3202 | 0.7300 | 0.2763 | 0.3386 | 0.2225 | – | 0.6646 | 0.5901 | 0.0877 | 0.0575 | 0.0059 |
| DFCN | 0.2264 | 0.3041 | 0.7276 | 0.3360 | 0.3241 | 0.0718 | – | 0.6593 | 0.5988 | 0.1516 | 0.0812 | 0.0152 |
| ♯best | 0.2937 | 0.4680 | 0.7124 | 0.4398 | 0.4527 | 0.2956 | – | 0.7263 | 0.6869 | 0.3161 | 0.0978 | 0.0563 |
| DGClu | 0.3306 | 0.3812 | 0.5073 | 0.1658 | 0.0569 | 0.1752 | – | 0.6098 | 0.5971 | 0.2043 | 0.0658 | 0.0107 |
| ♯ best | 0.3700 | 0.5145 | 0.7382 | 0.2913 | 0.2246 | 0.4023 | – | 0.6106 | 0.6041 | 0.2251 | 0.0821 | 0.0392 |
| AGC-DRR | 0.2630 | 0.4526 | 0.7445 | 0.4176 | 0.5425 | 0.2174 | – | 0.5679 | 0.4906 | 0.5688 | 0.1241 | – |
| ♯best | 0.2537 | 0.4626 | 0.7735 | 0.4305 | 0.5448 | 0.2487 | – | 0.5761 | 0.4973 | 0.6683 | 0.1614 | – |
| DCRN | 0.0836 | 0.4383 | 0.7737 | 0.4533 | 0.4226 | – | – | 0.0260 | 0.2948 | 0.2867 | 0.0118 | – |
| ♯best | 0.1014 | 0.5023 | 0.7738 | 0.4725 | 0.4444 | – | – | 0.0580 | 0.3000 | 0.6475 | 0.0665 | – |
| HSAN | 0.2639 | 0.4222 | 0.1025 | 0.2644 | 0.3752 | – | – | – | – | 0.2353 | – | – |
| ♯best | 0.3218 | 0.5737 | 0.5116 | 0.4051 | 0.4272 | – | – | – | – | 0.2572 | – | – |
| CCGC | 0.2432 | 0.3987 | 0.4642 | 0.3338 | 0.0165 | 0.0674 | – | 0.2428 | 0.2797 | 0.0638 | 0.0467 | 0.0451 |
| ♯best | 0.2885 | 0.5196 | 0.7059 | 0.4323 | 0.1864 | 0.2919 | – | 0.3310 | 0.4439 | 0.0924 | 0.0394 | 0.0571 |
| MAGI | 0.0739 | 0.5271 | 0.5525 | 0.4325 | 0.3748 | 0.1883 | 0.3225 | 0.4434 | 0.3301 | 0.1355 | 0.0868 | 0.0417 |
| ♯best | 0.3486 | 0.5353 | 0.7538 | 0.4483 | 0.4268 | 0.3089 | – | 0.5144 | 0.5547 | 0.2200 | 0.1108 | 0.0649 |
| NS4GC | 0.2568 | 0.5155 | 0.5166 | 0.4371 | 0.4963 | 0.3099 | – | 0.5834 | 0.4288 | 0.1154 | 0.1064 | 0.0268 |
| ♯best | 0.3463 | 0.5699 | 0.6976 | 0.4454 | 0.5314 | 0.3348 | – | 0.5931 | 0.4463 | 0.1738 | 0.1449 | 0.0509 |

Table 7: The F1 of 12 benchmark methods on 12 datasets.

| Method | Wiki | Cora | ACM | Citeseer | DBLP | PubMed | ARXIV | USPS | HHAR | Blog | Flickr | Roman |
|---|---|---|---|---|---|---|---|---|---|---|---|---|
| GAE | 0.3565 | 0.6165 | 0.8782 | 0.5431 | 0.4563 | 0.6478 | – | 0.5485 | 0.3640 | 0.4485 | 0.2424 | 0.1313 |
| ♯best | 0.4268 | 0.6909 | 0.8790 | 0.5506 | 0.4989 | 0.6513 | – | 0.5835 | 0.5549 | 0.4827 | 0.2766 | 0.1412 |
| GAE_S | 0.2827 | 0.6637 | 0.8475 | 0.5260 | 0.5008 | 0.6646 | – | 0.6376 | 0.3576 | 0.4386 | 0.1270 | 0.1288 |
| ♯best | 0.3235 | 0.6694 | 0.8509 | 0.5252 | 0.5090 | 0.6748 | – | 0.6376 | 0.3885 | 0.4385 | 0.2065 | 0.1317 |
| DAEGC | 0.2891 | 0.6354 | 0.8443 | 0.5699 | 0.5618 | 0.6540 | – | 0.6431 | 0.4802 | 0.3961 | 0.1634 | – |
| ♯best | 0.2960 | 0.6519 | 0.8963 | 0.5774 | 0.5780 | 0.6540 | – | 0.6445 | 0.4875 | 0.4222 | 0.1747 | – |
| SDCN | 0.0409 | 0.4422 | 0.8791 | 0.4546 | 0.6025 | 0.5529 | – | 0.7020 | 0.6141 | 0.1316 | 0.0823 | 0.0246 |
| ♯best | 0.0422 | 0.4598 | 0.8995 | 0.4540 | 0.6118 | 0.6038 | – | 0.7023 | 0.6271 | 0.2145 | 0.1255 | 0.0262 |
| DFCN | 0.3684 | 0.4620 | 0.8996 | 0.5268 | 0.5970 | 0.4270 | – | 0.6710 | 0.6226 | 0.3637 | 0.2907 | 0.1035 |
| ♯best | 0.4085 | 0.6812 | 0.8937 | 0.6372 | 0.7550 | 0.6722 | – | 0.7495 | 0.7643 | 0.5954 | 0.2991 | 0.1649 |
| DGClu | 0.4056 | 0.4711 | 0.7569 | 0.3819 | 0.3175 | 0.5802 | – | 0.6991 | 0.6293 | 0.4025 | 0.1236 | 0.1027 |
| ♯ best | 0.4239 | 0.6687 | 0.9047 | 0.5209 | 0.5795 | 0.7436 | – | 0.7092 | 0.6432 | 0.4213 | 0.1831 | 0.1149 |
| AGC-DRR | 0.3515 | 0.6378 | 0.9069 | 0.6213 | 0.7921 | 0.6291 | – | 0.6507 | 0.6439 | 0.7138 | 0.2586 | – |
| ♯best | 0.3554 | 0.6598 | 0.9194 | 0.6361 | 0.7962 | 0.6442 | – | 0.6692 | 0.6512 | 0.8288 | 0.3033 | – |
| DCRN | 0.2534 | 0.6043 | 0.9181 | 0.6400 | 0.6929 | – | – | 0.2237 | 0.2614 | 0.5307 | 0.1277 | – |
| ♯best | 0.2875 | 0.6802 | 0.9186 | 0.6554 | 0.7347 | – | – | 0.2464 | 0.2936 | 0.8371 | 0.2641 | – |
| HSAN | 0.3979 | 0.6252 | 0.5170 | 0.5027 | 0.6657 | – | – | – | – | 0.4402 | – | – |
| ♯best | 0.4360 | 0.7585 | 0.8079 | 0.6241 | 0.7114 | – | – | – | – | 0.4746 | – | – |
| CCGC | 0.3694 | 0.5583 | 0.6767 | 0.5486 | 0.2768 | 0.4222 | – | 0.3454 | 0.3075 | 0.2744 | 0.1442 | 0.1304 |
| ♯best | 0.4287 | 0.6976 | 0.8908 | 0.6152 | 0.5510 | 0.6497 | – | 0.5674 | 0.5006 | 0.3431 | 0.1701 | 0.1477 |
| MAGI | 0.1349 | 0.7179 | 0.8353 | 0.6242 | 0.6689 | 0.6155 | 0.2686 | 0.4718 | 0.3780 | 0.4001 | 0.3095 | 0.1004 |
| ♯best | 0.4172 | 0.7215 | 0.9106 | 0.6392 | 0.6991 | 0.6847 | – | 0.6553 | 0.6754 | 0.4878 | 0.3283 | 0.1605 |
| NS4GC | 0.3832 | 0.6787 | 0.7922 | 0.6242 | 0.7704 | 0.6793 | – | 0.6872 | 0.6018 | 0.3322 | 0.3120 | 0.1280 |
| ♯best | 0.4387 | 0.7441 | 0.8881 | 0.6399 | 0.7877 | 0.6956 | – | 0.7309 | 0.6425 | 0.4111 | 0.3727 | 0.1450 |

Table 8: The HOM of 12 benchmark methods on 12 datasets.

| Method | Wiki | Cora | ACM | Citeseer | DBLP | PubMed | ARXIV | USPS | HHAR | Blog | Flickr | Roman |
|---|---|---|---|---|---|---|---|---|---|---|---|---|
| GAE | 0.4655 | 0.5192 | 0.6185 | 0.3215 | 0.1896 | 0.2345 | – | 0.5234 | 0.4089 | 0.2964 | 0.1504 | 0.1002 |
| ♯best | 0.4987 | 0.5423 | 0.6206 | 0.3198 | 0.1909 | 0.2406 | – | 0.5222 | 0.5786 | 0.3228 | 0.1639 | 0.1041 |
| GAE_S | 0.3640 | 0.5376 | 0.6034 | 0.3078 | 0.2158 | 0.2502 | – | 0.5912 | 0.4527 | 0.2768 | 0.0880 | 0.0905 |
| ♯best | 0.4263 | 0.5472 | 0.6063 | 0.3089 | 0.2221 | 0.2610 | – | 0.5911 | 0.4619 | 0.2992 | 0.1206 | 0.0928 |
| DAEGC | 0.3262 | 0.4941 | 0.5175 | 0.3359 | 0.2291 | 0.2466 | – | 0.6586 | 0.5207 | 0.2333 | 0.1007 | – |
| ♯best | 0.3274 | 0.5211 | 0.6544 | 0.3517 | 0.2450 | 0.2466 | – | 0.6566 | 0.5276 | 0.2681 | 0.1048 | – |
| SDCN | 0.0293 | 0.3187 | 0.6342 | 0.2838 | 0.3143 | 0.1963 | – | 0.7460 | 0.6633 | 0.0518 | 0.0440 | 0.0111 |
| ♯best | 0.0316 | 0.3576 | 0.6729 | 0.2831 | 0.3305 | 0.2255 | – | 0.7400 | 0.6622 | 0.1026 | 0.0833 | 0.0104 |
| DFCN | 0.4354 | 0.4106 | 0.6712 | 0.3615 | 0.3180 | 0.0721 | – | 0.7324 | 0.6997 | 0.2567 | 0.1639 | 0.0633 |
| ♯best | 0.4587 | 0.5326 | 0.6565 | 0.4249 | 0.4353 | 0.3044 | – | 0.7534 | 0.7560 | 0.3787 | 0.1686 | 0.1321 |
| DGClu | 0.4538 | 0.4526 | 0.5027 | 0.2105 | 0.0736 | 0.2043 | – | 0.6899 | 0.6856 | 0.2533 | 0.0854 | 0.0390 |
| ♯ best | 0.4806 | 0.5510 | 0.6857 | 0.2911 | 0.2263 | 0.3409 | – | 0.6767 | 0.6900 | 0.2670 | 0.1130 | 0.0566 |
| AGC-DRR | 0.4120 | 0.5234 | 0.6825 | 0.4075 | 0.4889 | 0.2399 | – | 0.6762 | 0.6009 | 0.5719 | 0.2372 | – |
| ♯best | 0.4108 | 0.5300 | 0.7083 | 0.4177 | 0.4936 | 0.2576 | – | 0.6806 | 0.6011 | 0.6509 | 0.2439 | – |
| DCRN | 0.2530 | 0.5307 | 0.7156 | 0.4415 | 0.4055 | – | – | 0.1640 | 0.3845 | 0.3815 | 0.0608 | – |
| ♯best | 0.2792 | 0.5495 | 0.7156 | 0.4527 | 0.4338 | – | – | 0.2103 | 0.3861 | 0.6632 | 0.1677 | – |
| HSAN | 0.4748 | 0.5279 | 0.1982 | 0.3394 | 0.3893 | – | – | – | – | 0.3034 | – | – |
| ♯best | 0.4974 | 0.5927 | 0.4955 | 0.4129 | 0.4243 | – | – | – | – | 0.3142 | – | – |
| CCGC | 0.4517 | 0.4965 | 0.4571 | 0.3510 | 0.0434 | 0.0873 | – | 0.3569 | 0.3720 | 0.1040 | 0.0691 | 0.0947 |
| ♯best | 0.4819 | 0.5634 | 0.6473 | 0.4187 | 0.2323 | 0.3007 | – | 0.4810 | 0.5276 | 0.1447 | 0.0819 | 0.1147 |
| MAGI | 0.1848 | 0.5636 | 0.5272 | 0.4268 | 0.3765 | 0.1939 | 0.5061 | 0.5102 | 0.4098 | 0.2250 | 0.1548 | 0.0647 |
| ♯best | 0.5026 | 0.5673 | 0.6963 | 0.4388 | 0.4059 | 0.3019 | – | 0.5878 | 0.6415 | 0.2859 | 0.1784 | 0.1243 |
| NS4GC | 0.4483 | 0.5746 | 0.4839 | 0.4334 | 0.4561 | 0.3131 | – | 0.6696 | 0.5547 | 0.1914 | 0.1654 | 0.0846 |
| ♯best | 0.5245 | 0.6010 | 0.6356 | 0.4358 | 0.4840 | 0.3276 | – | 0.6724 | 0.5724 | 0.2490 | 0.2151 | 0.1197 |

Table 9: The COM of 12 benchmark methods on 12 datasets.

| Method | Wiki | Cora | ACM | Citeseer | DBLP | PubMed | ARXIV | USPS | HHAR | Blog | Flickr | Roman |
|---|---|---|---|---|---|---|---|---|---|---|---|---|
| GAE | 0.4326 | 0.5105 | 0.6225 | 0.3206 | 0.1918 | 0.2281 | – | 0.5269 | 0.6770 | 0.3071 | 0.1704 | 0.0932 |
| ♯best | 0.4536 | 0.5328 | 0.6246 | 0.3189 | 0.1941 | 0.2355 | – | 0.5270 | 0.6258 | 0.3392 | 0.1886 | 0.1614 |
| GAE_S | 0.5326 | 0.5302 | 0.6102 | 0.3061 | 0.2182 | 0.2437 | – | 0.5904 | 0.5625 | 0.2886 | 0.2199 | 0.0833 |
| ♯best | 0.4928 | 0.5391 | 0.6186 | 0.3093 | 0.2241 | 0.2583 | – | 0.5903 | 0.5909 | 0.2992 | 0.1431 | 0.0874 |
| DAEGC | 0.3417 | 0.4770 | 0.5187 | 0.3330 | 0.2372 | 0.2400 | – | 0.6650 | 0.7433 | 0.2393 | 0.1624 | – |
| ♯best | 0.3452 | 0.5074 | 0.6566 | 0.3487 | 0.2493 | 0.2400 | – | 0.6629 | 0.7428 | 0.2797 | 0.1599 | – |
| SDCN | 0.9313 | 0.3165 | 0.6405 | 0.3234 | 0.3217 | 0.1959 | – | 0.7513 | 0.7536 | 0.2954 | 0.1661 | 0.8262 |
| ♯best | 0.9364 | 0.3819 | 0.6758 | 0.3223 | 0.3449 | 0.2389 | – | 0.7482 | 0.7464 | 0.1975 | 0.1888 | 0.8166 |
| DFCN | 0.4246 | 0.4013 | 0.6730 | 0.3644 | 0.3289 | 0.0749 | – | 0.7698 | 0.7657 | 0.2802 | 0.2148 | 0.0616 |
| ♯best | 0.4583 | 0.5227 | 0.6585 | 0.4251 | 0.4417 | 0.3130 | – | 0.7794 | 0.7920 | 0.4065 | 0.2197 | 0.1328 |
| DGClu | 0.4616 | 0.5180 | 0.5470 | 0.2874 | 0.1108 | 0.2126 | – | 0.6918 | 0.7764 | 0.2957 | 0.1767 | 0.0360 |
| ♯ best | 0.4876 | 0.5680 | 0.6888 | 0.3154 | 0.2442 | 0.3452 | – | 0.6805 | 0.7798 | 0.3103 | 0.1655 | 0.0593 |
| AGC-DRR | 0.3962 | 0.5075 | 0.6846 | 0.4021 | 0.4886 | 0.2345 | – | 0.6735 | 0.6105 | 0.6172 | 0.3715 | – |
| ♯best | 0.3899 | 0.5125 | 0.7089 | 0.4127 | 0.4948 | 0.2521 | – | 0.6777 | 0.6127 | 0.6587 | 0.3199 | – |
| DCRN | 0.3307 | 0.5196 | 0.7175 | 0.4407 | 0.4101 | – | – | 0.3513 | 0.6035 | 0.4801 | 0.1699 | – |
| ♯best | 0.3933 | 0.5497 | 0.7168 | 0.4548 | 0.4341 | – | – | 0.3974 | 0.6006 | 0.6736 | 0.2685 | – |
| HSAN | 0.4339 | 0.5210 | 0.2871 | 0.3578 | 0.3878 | – | – | – | – | 0.3164 | – | – |
| ♯best | 0.4605 | 0.5897 | 0.5075 | 0.4171 | 0.4217 | – | – | – | – | 0.3266 | – | – |
| CCGC | 0.4597 | 0.5237 | 0.5224 | 0.3709 | 0.0607 | 0.0986 | – | 0.3673 | 0.4813 | 0.1158 | 0.0831 | 0.0918 |
| ♯best | 0.4832 | 0.5650 | 0.6495 | 0.4491 | 0.2425 | 0.3345 | – | 0.4894 | 0.5678 | 0.1594 | 0.0941 | 0.1124 |
| MAGI | 0.5262 | 0.5515 | 0.5420 | 0.4403 | 0.3824 | 0.1887 | 0.4359 | 0.5735 | 0.5250 | 0.2676 | 0.1669 | 0.0749 |
| ♯best | 0.5083 | 0.5567 | 0.6982 | 0.4487 | 0.4092 | 0.3029 | – | 0.5850 | 0.6710 | 0.3243 | 0.2018 | 0.1198 |
| NS4GC | 0.4063 | 0.5620 | 0.4904 | 0.4330 | 0.4528 | 0.3125 | – | 0.6680 | 0.5630 | 0.2278 | 0.1713 | 0.0773 |
| ♯best | 0.4740 | 0.5914 | 0.6378 | 0.4345 | 0.4806 | 0.3292 | – | 0.6714 | 0.5839 | 0.2748 | 0.2248 | 0.1110 |

Table 10: The SC of 12 benchmark methods on 12 datasets.

| Method | Wiki | Cora | ACM | Citeseer | DBLP | PubMed | ARXIV | USPS | HHAR | Blog | Flickr | Roman |
|---|---|---|---|---|---|---|---|---|---|---|---|---|
| GAE | 0.3029 | 0.2669 | 0.3972 | 0.2296 | 0.1628 | 0.3474 | – | 0.2548 | 0.3618 | 0.3708 | 0.3790 | 0.1671 |
| ♯best | 0.2727 | 0.2814 | 0.3975 | 0.2187 | 0.1476 | 0.3207 | – | 0.2509 | 0.2260 | 0.3633 | 0.1563 | 0.1614 |
| GAE_S | 0.2204 | 0.3802 | 0.4330 | 0.2475 | 0.1797 | 0.3629 | – | 0.2987 | 0.4990 | 0.3903 | 0.4142 | 0.1571 |
| ♯best | 0.2495 | 0.3471 | 0.4314 | 0.2263 | 0.1837 | 0.3501 | – | 0.2986 | 0.4451 | 0.3704 | 0.3929 | 0.1855 |
| DAEGC | 0.5434 | 0.3502 | 0.2835 | 0.2778 | 0.1135 | 0.2791 | – | 0.3853 | 0.3926 | 0.2809 | 0.2234 | – |
| ♯best | 0.5624 | 0.3304 | 0.4702 | 0.2488 | 0.1209 | 0.2791 | – | 0.3846 | 0.4003 | 0.3030 | 0.2689 | – |
| SDCN | −0.0001 | 0.2193 | 0.6336 | 0.1978 | 0.4865 | 0.5125 | – | 0.5401 | 0.4187 | 0.1715 | 0.0304 | −0.0151 |
| ♯best | −0.0222 | 0.2134 | 0.6261 | 0.1835 | 0.4011 | 0.3750 | – | 0.5016 | 0.3778 | 0.0145 | −0.1077 | −0.0098 |
| DFCN | 0.1827 | 0.2921 | 0.3513 | 0.3583 | 0.4423 | 0.5609 | – | 0.7458 | 0.6501 | 0.3390 | 0.1038 | 0.1675 |
| ♯best | 0.2124 | 0.1992 | 0.3020 | 0.1917 | 0.1055 | 0.2906 | – | 0.6827 | 0.4788 | 0.2323 | 0.1092 | 0.1235 |
| DGClu | 0.6032 | 0.3221 | 0.3739 | 0.1646 | 0.0716 | 0.2352 | – | 0.6087 | 0.8652 | 0.5872 | 0.4440 | 0.5378 |
| ♯ best | 0.5993 | 0.4240 | 0.4117 | 0.1927 | 0.1276 | 0.2526 | – | 0.5878 | 0.8814 | 0.5650 | 0.3627 | 0.2868 |
| AGC-DRR | 0.4076 | 0.5586 | 0.7683 | 0.4806 | 0.6489 | 0.7005 | – | 0.6076 | 0.6160 | 0.5662 | 0.2398 | – |
| ♯best | 0.3907 | 0.5257 | 0.6547 | 0.4668 | 0.6139 | 0.6333 | – | 0.6009 | 0.5839 | 0.4784 | 0.0723 | – |
| DCRN | 0.1673 | 0.3020 | 0.4586 | 0.4032 | 0.3938 | – | – | 0.4402 | 0.2035 | 0.0494 | 0.4127 | – |
| ♯best | 0.1787 | 0.2086 | 0.4422 | 0.2691 | 0.2553 | – | – | 0.2585 | 0.1266 | 0.1645 | 0.1809 | – |
| HSAN | 0.2115 | 0.1002 | 0.0492 | 0.0427 | 0.1986 | – | – | – | – | 0.3379 | – | – |
| ♯best | 0.1994 | 0.1098 | 0.0735 | 0.0482 | 0.1914 | – | – | – | – | 0.2920 | – | – |
| CCGC | 0.2625 | 0.1654 | 0.1893 | 0.2252 | 0.2474 | 0.0990 | – | 0.2347 | 0.4298 | 0.1804 | 0.2815 | 0.1358 |
| ♯best | 0.2050 | 0.0887 | 0.0648 | 0.0615 | 0.0232 | 0.1140 | – | 0.2259 | 0.2688 | 0.1537 | 0.2956 | 0.1090 |
| MAGI | 0.0273 | 0.1888 | 0.6203 | 0.3386 | 0.2834 | – | 0.2951 | 0.5314 | 0.5630 | 0.3777 | 0.2679 | 0.2812 |
| ♯best | 0.3900 | 0.1893 | 0.6598 | 0.2905 | 0.4028 | 0.4048 | – | 0.1979 | 0.2065 | 0.2533 | 0.1810 | 0.1642 |
| NS4GC | 0.2921 | 0.2645 | 0.1502 | 0.1386 | 0.1275 | 0.2219 | – | 0.3199 | 0.4499 | 0.1591 | 0.1173 | 0.1104 |
| ♯best | 0.2804 | 0.2689 | 0.3135 | 0.1346 | 0.1435 | 0.2272 | – | 0.3181 | 0.4405 | 0.1661 | 0.1815 | 0.1339 |

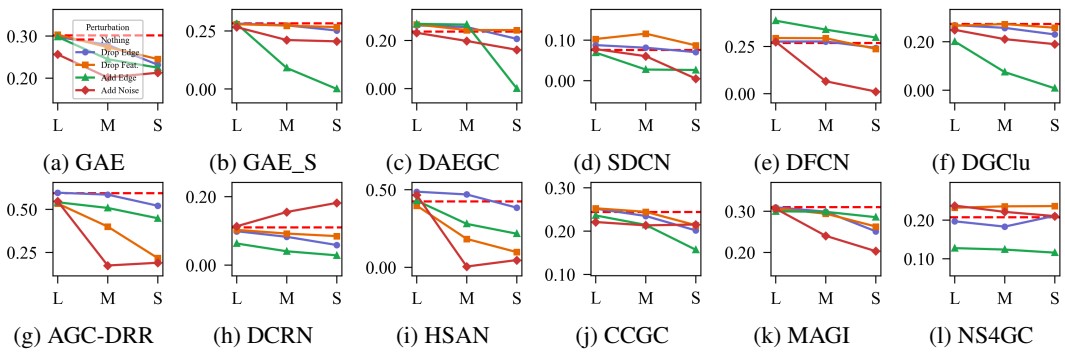

(a) GAE  (b) GAE_S  (c) DAEGC  (d) SDCN  (e) DFCN  (f) DGClu

(g) AGC-DRR  (h) DCRN  (i) HSAN  (j) CCGC  (k) MAGI  (l) NS4GC

Figure 6: NMI of 12 methods on Blog with light (L), moderate (M), and severe (S) perturbations.

suffered significant performance degradation under moderate to severe perturbations, while light perturbations even improved performance for some methods. To further investigate their robustness on the dataset with a low homophily ratio, we conducted experiments on the Blog dataset, with results shown in Figure 6. Overall, the clustering performance of all methods declined as perturbation intensity increased, indicating their insufficient robustness to perturbations. Analysis of perturbation type impacts revealed that edge addition had a more

significant effect on method performance, likely due to reduced graph homogeneity caused by increased edges. Notably, certain methods like DAEGC, SDCN, and HSAN showed improved performance after perturbations, which offers new insights for subsequent robustness improvements.

## B.3 Additional Visualization Results

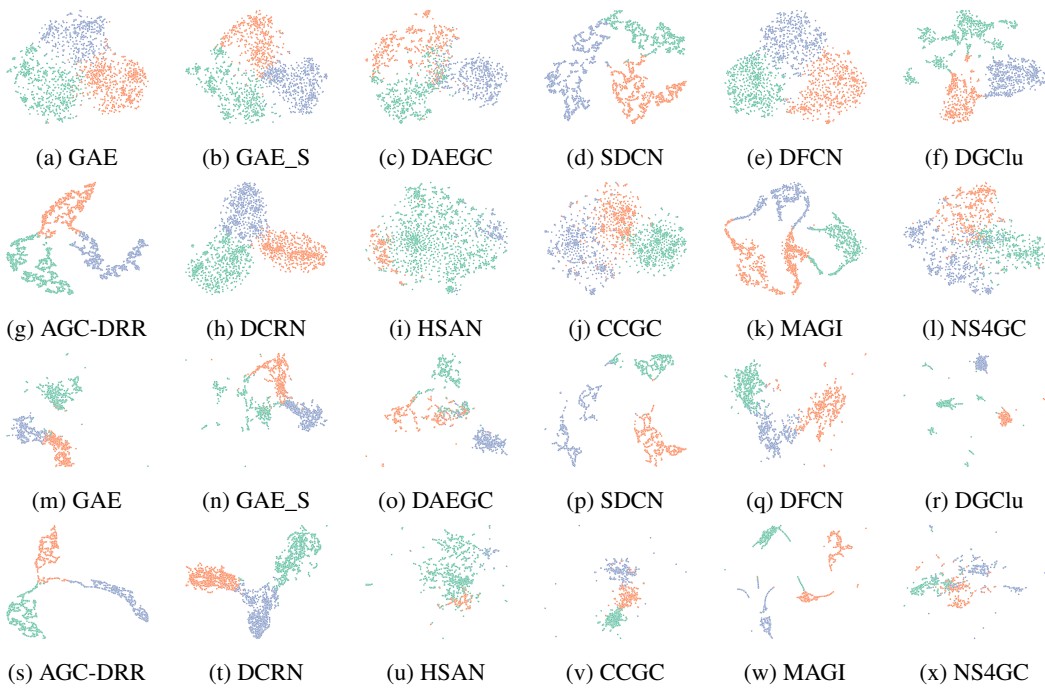

| (a) GAE | (b) GAE_S | (c) DAEGC | (d) SDCN | (e) DFCN | (f) DGClu |
|---------|-----------|-----------|----------|----------|-----------|
| (g) AGC-DRR | (h) DCRN | (i) HSAN | (j) CCGC | (k) MAGI | (l) NS4GC |
| (m) GAE | (n) GAE_S | (o) DAEGC | (p) SDCN | (q) DFCN | (r) DGClu |
| (s) AGC-DRR | (t) DCRN | (u) HSAN | (v) CCGC | (w) MAGI | (x) NS4GC |

Figure 7: $t$-SNE (first and second rows) and UMAP (third and fourth rows) 2D visualization of embeddings learned by 12 benchmark algorithms on ACM.

## C Open-source Package

We provide an open-source Python package named PyDGC[3], which adopts a modular architecture. The **datasets** module not only includes 12 benchmark datasets but also is compatible with other data from the PyG and OGB ecosystems. The **models** module has 12 representative Benchmark models built in, covering a variety of research paradigms. The **clusterings** module provides commonly used clustering algorithms. The **metrics** module implements the calculation of the core indicators of the evaluation system. The **utils** module offers common utility functions for configuration, logging, device, data transformation, visualization, and more. In addition, the library is designed with a **DGCModel** class and a standard pipeline interface **BasePipeline**. This ensures the reproducibility of the built-in methods and also makes it convenient for developers to expand new algorithms. We will continue to maintain the built-in methods and construct a leaderboard. We hope to build the DGC ecosystem with everyone jointly.

Here is an example demonstrating how to quickly reproduce clustering of GAE on the Cora dataset using PyDGC:

```
1  from pydgc.utils import parse_arguments
2  from pydgc.pipelines import import GAEPipeline
3
4  args = parse_arguments("CORA")
5  pipeline = GAEPipeline(args)
6  pipeline.run()
```

---

[3]https://github.com/Marigoldwu/PyDGC

# D Parameter Settings

Twelve benchmark methods involve varying numbers of hyperparameters, and some require pretraining with parameter configurations different from those in the formal training phase. To ensure fair comparison of their true performance in unsupervised clustering tasks, parameter settings adopt the following strategy: if the original literature provides optimized parameters for specific datasets, directly use the configurations stated in the literature; if dataset-specific parameters are not provided, set them according to the publicly available hyperparameter analysis strategies of the methods; if neither condition is met, use default parameters in the source code. Specific parameter configurations are detailed in Table 11.

Table 11: Default hyper-parameters for benchmark algorithms.

| | **Pretrain** | **Train** |
|---|---|---|
| GAE | / | lr=1e-3, max_epoch=30 |
| GAE_SSC | lr=1e-3, max_epoch=30 | lr=1e-3, max_epoch=20, $\alpha$=1e-3 |
| DAEGC | lr=1e-3, max_epoch=30, weight_decay=1e-3, $t$=2 | lr=1e-3, max_epoch=100, weight_decay=1e-3, $\gamma$=10, $t$=2 |
| SDCN | lr=1e-3, max_epoch=30 | lr=1e-3, max_epoch=50 or 200, $\alpha$=0.1, $\beta$=0.01, $\sigma$=0.5 |
| DFCN | lr=1e-3, max_epoch=30 or 100, $\gamma$=0.1, $\alpha$=0.1, $\beta$=0.01, $\omega$=0.1, PCA=100 | lr=1e-4, max_epoch=200, $\gamma$=0.1, $\lambda$=10, PCA=100 |
| DCRN | lr=1e-3, max_epoch=30 or 100, $\gamma$=0.1, $\alpha$=0.1, $\beta$=0.01, $\omega$=0.1, PCA=100, alpha_value=0.2 | lr=1e-4, max_epoch=200, $\gamma$=1000, $\lambda$=10, PCA=100, alpha_value=0.2 |
| AGC-DRR | / | lr=1e-4, view_lr=1e-3, max_epoch=200, reg_lambda=1 |
| CCGC | / | lr=1e-4, max_epoch=400, $\tau$=0.5, $t$=4, $\alpha$=0.5 |
| HSAN | / | lr=1e-3, max_epoch=400, $\beta$=1, $\tau$=0.9, t=2 |
| DGClu | / | lr=1e-3, max_epoch=300, $\alpha = 0.0$, $\lambda = 0.2$ |
| MAGI | / | lr=5e-4, max_epoch=400, weight_decay=1e-3, $\tau$=0.5, $\alpha$=0.5 |
| NS4GC | / | lr=1e-3, max_epoch=200, weight_decay=1e-5, $\lambda = 1.0$, $\gamma = 1.0$, s=0.6, $\tau = 0.1$, $p_{d_1} = p_{d_2} = 0.5$, $p_{m_1} = p_{m_2} = 0.1$ |

