# OpenReview forum: "DGCBench:  A Deep Graph Clustering Benchmark"
_NeurIPS.cc/2025/Datasets_and_Benchmarks_Track — NeurIPS 2025 Datasets and Benchmarks Track poster_

### Official Review · Reviewer_oT3N · 2025-06-09

**Rating:** 4
**Confidence:** 4

**Summary:**

This paper addresses the problem of graph clustering benchmarks by selecting appropriate datasets, organizing evaluation perspectives, and conducting detailed evaluation experiments.

**Additional Feedback:**

**F1.** In general, it is preferable not to use colored fonts in academic papers.

**F2.** In Section 4.1, the term "generative methods" appears for the first time, but it is unclear which specific existing methods this refers to.

**Dataset Code Accessibility:**

Yes

**Dataset Code Comments:**

- The code is publicly available on GitHub and has already received 110 stars.

**Ethical Considerations:**

No, there are no or only very minor ethics concerns

**Final Justification:**

I agree that the author's responses are carefully made, and the proposal made a good enough contribution as a Graph Clustering Benchmark. I raised the score 1 point up.

**Limitations Weaknesses:**

**L1.** The 12 benchmark datasets are not particularly novel. The large-scale datasets, Pubmed, Roman, and ARXIV, are all commonly used in the GNN community.

**L2.** Except for the distinction between final and intermediate results, the six evaluation dimensions in the benchmark do not offer distinctive insights. Moreover, discriminability is typically evaluated using the silhouette coefficient, and visualization-based analysis is inadequate for quantitative assessment.

**L3.** In Section 4.1, the paper states: "Analysis of homogeneous graphs shows that ... the optimal clustering values of all methods do not significantly exceed the inherent homophily ratio of the datasets." However, since clustering values (ACC, NMI) and the homophily ratio have different units, discussing whether one exceeds the other in absolute value is not meaningful: only correlation should be discussed.

**L4.** The training curves of loss and NMI shown in Figure 2 offer an interesting view into their correlation, but the observations appear highly method-dependent, making further interpretation difficult.

**L5.** RQ1 and RQ2 are interesting, but the remaining research questions are rather generic and lack novelty.

**Strengths Contributions:**

**S1.** DGCBench is a comprehensive and unified benchmark designed specifically for deep graph clustering methods, making it suitable for the *Dataset & Benchmark Track*.

**S2.** By using DGCBench, the authors evaluated 12 state-of-the-art DGC methods on 12 datasets spanning various domains and scales, across six critical dimensions: discriminability, effectiveness, scalability, efficiency, stability, and robustness.

**S3.** The evaluation perspective that distinguishes between *final results (algorithm-terminated outputs)* and *best intermediate results (optimal solutions identified during training)* is a novel and interesting contribution.

**S4.** The paper presents exhaustive evaluation experiments based on six research questions.

---

> ### Author Rebuttal · Authors · 2025-07-27
>
> Thank you sincerely for the constructive comments. We clarify your concerns point by point:
>
> ### **Datasets Novelty (L1)**
>
> 1. DGCBench aims to form a **unified and representative** benchmark dataset. Therefore, we place more emphasis on the datasets' coverage and representativeness. This has been recognized by all reviewers.
>
> 2. As you noted, datasets such as PubMed, Roman, and ARXIV are widely used in the GNN community. **Nevertheless, in the DGC field, Roman and ARXIV are rarely used, and only MAGI (Liu et al., KDD2024) can cluster ARXIV among the 12 baseline methods**. Roman and ARXIV represent **heterogeneous** data and **large-scale** data, respectively. As they are widely used in the GNN community, using both datasets to evaluate benchmark methods **ensures better fairness and reproducibility of the benchmark**.
>
> 3. We will supplement biomedical data for further benchmarking and actively advance this. We have reserved interfaces for expanding datasets in our toolkit.
>
> ### **Distinctive Insights (L2-1)**
>
> Besides the insights of 'homophily bottleneck' and 'distinction between final and intermediate results', the following insights are also valuable:
>
> - **Stability**. We have identified counterintuitive phenomena in specific methods, namely, the simultaneous increase of loss and performance. We speculate this is caused by the conflict between the feature learning loss and self-supervised clustering loss, which needs further research.
>
> - **Robustness**. We have found that some 'noises' can even improve performance. This violates the intuition  'noises inevitably impair performance'; the underlying mechanism urgently needs to be analyzed.
>
> ### **Quantitative Assessment of Discriminability (L2-2)**
>
> Besides visualizations, **Table 5 has already presented three quantitative metrics** (completeness, homogeneity, and silhouette coefficient) for 12 methods on the Cora dataset. The discriminability reflected by these quantitative metrics is highly consistent with the visualization results. In the appendix, we already provided the complete silhouette coefficient of 12 methods on 12 datasets in Table 10. In addition, we have obtained the visualization results on the ACM dataset to further illustrate the poor discriminability of embeddings learned by these methods. We will add them to the appendix later.
>
> ### **Correlation between Homophily Ratio and Clustering Metrics (L3)**
>
> Thank you for this professional suggestion. **The experimental results demonstrate that the inherent homophilic structure exerts a significant impact on the upper bound of clustering performance on the high homophily graphs.** This correlation **underscores the 'homophily bottleneck'** in current DGC methods, **emphasizing the need for new paradigms** (e.g., beyond GNNs or novel self-supervised objectives) to overcome this limitation. We will clarify this correlation in the final version as follows.
>
> Experimental results indicate that on most high-homophily datasets, the performance of DGC algorithms closely approximates the inherent homophily ratios of datasets. Moreover, no significant performance improvements are observed on some datasets (e.g., DBLP, Cora, and Citeseer). This phenomenon suggests that intrinsic homophily may constitute a potential upper bound for clustering performance, where clustering results can hardly surpass the level determined by the homophilic structure of data.
>
> ### **Interpretation of Counterintuitive Phenomena between Loss and NMI (L4)**
>
> It is indeed difficult to provide a unified interpretation, as different methods have different designs. This observation primarily serves to illustrate that **the loss functions used to train neural networks in some methods may be disconnected from the clustering objectives**, which provides important insights for the design of clustering loss functions.
>
> ### **Research Question Novelty (L5)**
>
> We appreciate your positive evaluation of RQ1 and RQ2. We acknowledge that, when viewed individually, any single evaluation dimension may not be highly innovative, but such evaluations are mainstream. In the context of DGC, our evaluation system is **comprehensive** and addresses the shortcomings of previous evaluations. This has also been recognized by all reviewers and is precisely the novel aspect of our evaluation.
>
> ### **Fonts Colors (F1)**
>
> Thank you! In the text, we used colors to highlight the six different evaluation dimensions. In the tables, colors were used to highlight methods with different rankings, and background color intuitively denotes the standard deviation level. These styles are also used in other benchmarks (GSLB, Li et al., NeurIPS 2023; Opengsl, Zhou et al., NeurIPS 2023).
>
> ### **"Generative methods" Explanation (F2)**
>
> Thank you for noting this ambiguity. 'Generative methods' in our manuscript specifically refer to methods under the reconstructive paradigm, including GAE, GAE\_S, DAEGC, SDCN, and DFCN, which learn embeddings by reconstructing graph structure or features. We will clarify 'generative methods' in the final version.

---

> > ### Comment · Reviewer_oT3N · 2025-08-08
> >
> > I agree that the author's responses are carefully made, and the proposal made a good enough contribution as a Graph Clustering Benchmark. I raised the score 1 point up.

---

> ### Author Response · Authors · 2025-08-05
>
> Dear Reviewer oT3N,
> We noticed that we haven’t received your feedback on our rebuttal. We understand you might be busy with other reviews, and any brief response would be very helpful for us to better address your concerns. Should you have no further concerns, we respectfully request you to reconsider the scoring. Thank you for your time and consideration.
> Best regards,

---

### Official Review · Reviewer_7jij · 2025-06-12

**Rating:** 5
**Confidence:** 4

**Summary:**

The authors present the first benchmark for deep graph clustering, including 12 datasets with different characteristics (comprehensive scale coverage, diverse feature spaces, rich data type diversity), 12 baseline methods covering different paradigms (reconstructive, predictive, contrastive), and conducts a comprehensive experimental analysis from six dimensions such as discriminability, robustness, stability, and scalability. Through experimental analysis, several insights are proposed that contribute to the development of the community, including the homophily bottleneck, training instability, and vulnerability to perturbations. The authors also provide an easy-to-use open-source toolkit to facilitate the reproduction and development of DGC algorithms.

**Additional Feedback:**

1.	Please provide further explanations and clarifications on the future research directions. The current description of the future directions seems to focus solely on the issues at hand, without specifying exactly how to do it and what the possible solutions are.

2.	Why isn't hyperparameter search used in hyperparameter settings to obtain the optimal parameters? I am concerned about the fairness of the results because it is usually difficult to obtain the best model performance without using the most suitable hyperparameters.

3.	Since PyDGC is a Python package, it is best to provide interface documentation for further instructions.

4.	It is essential to clarify that the NMI in Table 4 should not include any units of measurement, and it must not be expressed as 'its/s'.

**Dataset Code Accessibility:**

Yes

**Dataset Code Comments:**

This project is open on GitHub with several stars.

**Ethical Considerations:**

No, there are no or only very minor ethics concerns

**Final Justification:**

Thanks to the authors for clarifying most of my concerns. I have read the explainations and expect the modification in the new version of this work.

**Limitations Weaknesses:**

1. Although the authors discussed the potential research directions of DGC, the description was rather vague. This vagueness weakens the article's ability to effectively guide and inspire future research efforts in the DGC field.

2. PyDGC lacks detailed code development documentation. Although the documentation provides guidance on reproducing the built-in algorithm and extending the new model, the lack of detailed parameter descriptions for related functions or interfaces increases the difficulty of implementation.

**Strengths Contributions:**

1.  This article is well-structured and easy to follow.

2. The proposed DGCBench covers a comprehensive range of algorithms, datasets, and evaluation dimensions. The open-source toolkit PyDGC facilitates the reproduction and expansion of algorithms.

3. Extensive experiments and analyses provided interesting phenomena and insightful conclusions of the state-of-the-art DGC algorithms. These findings serve as a solid foundation for future research, fostering  the continuous advancement of the DGC community.

---

> ### Author Rebuttal · Authors · 2025-07-27
>
> Thank you for your insightful comments and valuable suggestions. Below are our detailed responses to the raised points:
>
> ### **Future Research Directions (F1\&L1)**
>
> We fully agree that the description of future research directions can be more guiding and operable. In the final version, we will add specific exploration ideas and potential solutions to enhance their relevance and practicality:
>
> - **Homophily Bottleneck**. Although the homophily assumption enabled early GNN-based clustering methods to achieve better results than those that did not consider such relationships, this assumption now limits the upper bound of clustering performance. Future methods should rely less on the homophily assumption.
>
> - **Stability Deficiency**. Benchmark methods suffer the sensitivity to initial parameters and unstable convergence during the training process, which reduces the reliability of the algorithm and restricts its practical application. It is urgent to explore convergence methods with strict theoretical support.
>
> - **Robustness Gap**: The message passing mechanism of GNNs can accelerate the diffusion of noise, causing the model to be sensitive to severe noise. Future efforts can focus on noise source suppression, anti-noise aggregation, and result calibration.
>
> - **Efficiency Plateau**: Despite years of research, DGC methods have not achieved substantial progress in comprehensive efficiency.
>
> - **Scalability Challenges**: DGC faces challenges in scaling on large-scale datasets, with high computational and storage costs, which hinder its deployment in resource-constrained environments. Although sampling enables large-scale graph learning, how to balance global structures in sampling and how to implement large-scale clustering still need to be explored.
>
> - **Discriminability Limitations**: Although the learned embeddings exhibit strong intra-cluster cohesion, their inter-cluster discriminability is significantly poor. This indicates that future efforts should focus on the discriminative features learning of clustering and hard samples.
>
> ### **Hyperparameter Search (F2)**
>
> As deep graph clustering is an unsupervised task, hyperparameter search lacks ground-truth labels for validation, making 'optimal' parameters hard to objectively define. We followed standard practice that uses default parameters from original papers or domain-accepted ranges to ensure fair comparison. The hyperparameter settings in our research **were already listed in the appendix**.
>
> ### **PyDGC Documentation (F3\&L2)**
>
> Thank you for the suggestion. The API documentation has been uploaded to 'Read the Docs'.
>
> ### **Unit of Metrics (F4)**
>
> Thank you for the careful review. The 'its/s' label for NMI in Table 4 is a typesetting error. We will fix it in the final version.

---

### Official Review · Reviewer_VyTv · 2025-06-30

**Rating:** 5
**Confidence:** 4

**Summary:**

This paper presents DGCBench, a comprehensive and unified benchmarking framework for Deep Graph Clustering (DGC) methods. Recognizing the challenges posed by the unsupervised nature of DGC and the lack of standardized evaluation protocols, the authors evaluate 12 state-of-the-art DGC methods on 12 datasets from diverse domains. The evaluation spans six critical dimensions: discriminability, effectiveness, scalability, efficiency, stability, and robustness. To support reproducibility and standardization, the authors also introduce PyDGC, an open-source Python library that streamlines training and evaluation procedures. The study identifies several common limitations of existing approaches, including reliance on homophily assumptions, sensitivity to perturbations, and scalability bottlenecks, offering valuable insights for future research in the field.

**Additional Feedback:**

The literature review on deep graph clustering (DGC) methods could be more comprehensive.

**Dataset Code Accessibility:**

Yes

**Ethical Considerations:**

No, there are no or only very minor ethics concerns

**Final Justification:**

Thank you to the authors for the rebuttal. My concerns have been addressed, and I am maintaining my score of 'Accept'.

**Limitations Weaknesses:**

1. In Equation (1), some variables such as Z are in boldface (indicating vector or matrix form), while others like Y are not. This inconsistency may confuse readers regarding their intended dimensionality or type. Could the authors clarify whether this is a typesetting issue or a conceptual distinction?

2. In Tables 2 and 3, some numerical entries are highlighted with background colors while others are not, without clear explanation. This inconsistency may confuse readers and undermines the interpretability of the reported results.

3. The paper applies both t-SNE and UMAP for embedding visualization but lacks a discussion of their respective characteristics, potential differences in outcomes, or whether the observations generalize across other datasets.

**Strengths Contributions:**

1. The paper addresses the lack of standardized benchmarks for Deep Graph Clustering (DGC), an important yet underexplored task in graph representation learning.

2. DGCBench systematically evaluates 12 representative DGC methods across 12 diverse datasets, covering major paradigms and real-world scenarios.

3. The benchmark goes beyond accuracy and includes robustness, scalability, stability, and discriminability, offering a more holistic assessment of model performance.

4. The paper is well-structured and easy to follow, with informative figures and tables that support the analysis.

---

> ### Author Rebuttal · Authors · 2025-07-27
>
> Thank you for your valuable comments on improving our work. Our clarifications are below:
>
> ### **Symbol Style (L1)**
>
> The inconsistent formatting of variables in Eq. (1) is a typesetting oversight, not a deliberate conceptual distinction. $\mathbf{Z}$ is the embedding matrix and $\mathbf{Y}$ is the label vector. We will standardize variable formatting in the final version.
>
> ### **Meaning of Cell Shade (L2)**
>
> The cell shade indicates the standard deviation of results in 10 experiments, with darker colors representing larger standard deviations and lighter colors indicating smaller ones, which **was detailed in the caption of Table 2**. Besides, a legend at the bottom of Table 2 specifies the correspondence between color shades and their respective standard deviation ranges.
>
> ### **Visualization Methods (L3)**
>
> We agree that discussing the characteristics of visualization methods and differences in their outcomes is valuable, and we would like to supplement the relevant details as follows: **Both t-SNE and UMAP are used in this manuscript to intuitively demonstrate the spatial distribution characteristics of clustering results**. Their differences will be added in the revised version: t-SNE focuses more on preserving local structures, making it suitable for revealing fine-grained intra-cluster relationships. UMAP, while retaining local structures, also takes global topology into account, which is more conducive to presenting the overall distribution pattern between clusters. In the experiments, **the two methods show consistency** in the main conclusion: intra-cluster cohesion is high, while inter-cluster discriminability is poor. Regarding the generalization of the results, **we will add these results in the appendix**.
>
> ### **Literature Review (L4)**
>
> Thanks for your constructive comments. Due to space constraints, we only introduced representative methods in the current submission. Figure 1 sorted out the roadmap of some DGC methods and summarized them into three paradigms: reconstructive, predictive, and contrastive. The technical details of 12 benchmark algorithms were detailed in Appendix A. Additionally, **our open-source toolkit provides a more extensive list of method classifications and applications (including relevant papers and open-source code)**. To demonstrate a more comprehensive literature review, we will include the following works in the final version:
>
>   - **Reconstructive**: GRACE (Yang et al., IEEE BigData 2020), CaEGCN (Huo et al., IEEE TKDE 2021), AGCN (Peng et al., ACM MM 2022), GC-VGE (Guo and Dai, Pattern Recognition 2022), FT-VGAE (Mrabah et al., IJCAI 2022), R2FGC (Yi et al., IEEE TNNLS 2023), Hole (Gu et al., CIKM 2023), GC-SEE (Ding et al., Pattern Recognition 2023), DGCN (Pan and Kang, ICML 2023), UDGC (Jiao and Li, TNNLS 2024), FastDGC (Ding et al., ACM TKDD 2024)
>
>   - **Predictive**: MTEL (Zhang et al., Neural Networks 2022), MLGAL (Yu and Jia, Knowledge-based Systems 2023), DICN (Chai et al., Information Sciences 2024)
>
>   - **Contrastive**: GDCL (Zhao et al., IJCAI 2021), CGC (Park et al., Web Conf., 2022), MLG-CPC (Zhang et al., IJCAI 2023), SCGC (Liu et al., TNNLS 2023), CONGREGATE (Sun et al., IJCAI 2023), RGC (Liu et al., ACM MM 2023), CONVERT (Yang et al., ACM MM 2023), CLAGC (Zheng et al., Information Sciences 2024), GMIM (Ahmadi et al., Knowledge-based Systems 2024), HomoCAGC (Chen et al., IEEE TCSVT 2025), THESAURUS (Deng et al., AAAI 2025)

---

> ### Comment · Area_Chair_hqXD · 2025-08-05
>
> Dear Reviewer VyTv, Could you please read the authors’ responses to your reviews and comment on whether they have addressed your concerns? Please note that only submitting the acknowledgement without interaction will be considered an insufficient review.

---

> ### Comment · Reviewer_VyTv · 2025-08-06
>
> Thank you to the authors for the rebuttal. My concerns have been addressed, and I am maintaining my score of 'Accept'.

---

> > ### Author Response · Authors · 2025-08-06
> >
> > Thank you so much for taking the time to review our rebuttal and for your continued support of our work.

---

### Official Review · Reviewer_X2Wi · 2025-07-01

**Rating:** 5
**Confidence:** 5

**Summary:**

This paper presents DGCBench, a comprehensive benchmark suite for deep graph clustering (DGC). The authors integrate 12 widely used graph datasets, covering diverse graph structures and scales, and systematically evaluate 12 representative DGC methods from generative, predictive, and contrastive learning paradigms. The benchmark includes evaluation across six key dimensions: clustering performance, stability, robustness, efficiency, scalability, and discriminability. Furthermore, the authors release an open-source library, PyDGC, to facilitate fair comparisons and promote reproducibility in the DGC community.

**Dataset Code Accessibility:**

Yes

**Dataset Code Comments:**

The authors provide full access to their benchmark toolkit (PyDGC), ensuring reproducibility and usability:
Code Availability: The GitHub repository (https://github.com/Marigoldwu/PyDGC)
The GitHub repo is actively maintained (no "under development" disclaimers).

**Ethical Considerations:**

No, there are no or only very minor ethics concerns

**Final Justification:**

The authors have addressed my concerns. Taking into account the opinions of all the reviewers, I maintain my initial evaluation.

**Limitations Weaknesses:**

- Lack of theoretical analysis of observed phenomena (e.g., loss increases but clustering improves, homogeneity assumption leads to performance bottlenecks).
- Limited large-scale validation (e.g., clustering results of different methods in ARXIV).
- The dataset component lacks originality; all included datasets are publicly available, with no newly curated or domain-specific data introduced.
- The benchmark primarily focuses on existing methods and does not propose novel algorithms or technical innovations for DGC itself.

**Strengths Contributions:**

- The paper addresses a clear gap in the DGC field, where systematic, standardized benchmarks are lacking, making method comparisons difficult.
- The benchmark covers a wide range of datasets, including both homogeneous and heterogeneous graphs, ensuring broad applicability.
- Evaluation spans multiple important dimensions beyond accuracy, such as robustness and scalability, providing a holistic view of DGC methods.
- The authors provide a well-documented, user-friendly open-source library (PyDGC), which lowers the barrier to entry and encourages community adoption.

---

> ### Author Rebuttal · Authors · 2025-07-27
>
> Thank you for the valuable comments, and we clarify them as follows.
>
> ### **Theoretical Analysis (L1)**
>
> We agree with your valuable comment. Currently, the DGCBench benchmark primarily focuses on analysis through comprehensive empirical investigation. In the subsequent expansion efforts, we plan to incorporate theoretical content to further enhance its depth and systematicness.
>
> ### **Large-scale Validation (L2)**
>
> Tables 2 and 3 in the manuscript already present the clustering results on the PubMed (19,717 nodes), Roman (22,662 nodes), and ARXIV (169,343 nodes) datasets.  Since methods (except MAGI) use full batch and do not provide training strategies for mini batch, they all **have GPU memory overflow (24GB VRAM)  labeled with "-"**. These results objectively reflect the scalability challenges of existing methods. Section 4.5 briefly comments on the reasons behind the scalability of some methods.
>
> ### **Originality of Datasets (L3)**
>
> We aim to build unified and representative benchmark datasets covering diverse types. Therefore, we place greater emphasis on the coverage and representativeness, which has been recognized by all reviewers. Additionally, in the outlook of our manuscript and the documentation of open-source codes, we have mentioned that we will supplement biomedical data for further testing, and we are actively advancing this. We have reserved interfaces for expanding datasets in our toolkit.
>
> ### **No Novel Algorithm or Technique Proposed (L4)**
>
> The core focus of our work is to construct a standardized benchmark for the DGC field, and systematically evaluate existing methods rather than propose new clustering algorithms. Although no new algorithms are directly proposed, through comprehensive and systematic evaluations as well as in-depth analyses, **we have put forward several research-worthy questions and potential research directions, and pointed out possible solutions in the future outlook**, which we believe are of great significance to the DGC community, such as:
>
> - **Homophily Bottleneck**. Although the homophily assumption enabled early GNN-based clustering methods to achieve better results than those that did not consider such relationships, this assumption now limits the upper bound of clustering performance. Future methods should rely less on the homophily assumption.
>
> - **Stability Deficiency**. Benchmark methods suffer the sensitivity to initial parameters and unstable convergence during the training process, which reduces the reliability of the algorithm and restricts its practical application. It is urgent to explore convergence methods with strict theoretical support.
>
> - **Robustness Gap**: The message passing mechanism of GNNs can accelerate the diffusion of noise, causing the model to be sensitive to severe noise. Future efforts can focus on noise source suppression, anti-noise aggregation, and result calibration.

---

> ### Comment · Area_Chair_hqXD · 2025-08-01
>
> Dear Reviewer X2Wi, as per your suggested weakness, please note that the benchmark track does not have a novel algorithm requirement.

---

> ### Author Response · Authors · 2025-08-05
>
> Dear Reviewer X2Wi, thanks for your constructive feedback. We wanted to check in briefly about our rebuttal, and we’d value any brief thoughts you might have. Even a short note would help us ensure we’ve addressed your concerns effectively. We truly appreciate your time. Best regards,

---

### Decision · Program_Chairs · 2025-09-18

**Decision:**

Accept (poster)

**Comment:**

All reviewers agreed that the paper fills an important gap by providing the first comprehensive benchmark for deep graph clustering, accompanied by the open-source PyDGC library. The evaluation across 12 methods and 12 datasets, along multiple dimensions, was considered thorough and valuable for the community. The shared codebase is already popular on GitHub.

The main concerns raised were about the absence of theoretical analysis, limited novelty in datasets, and some clarity issues in presentation (notation, visualization, documentation). Authors clarified these points in the rebuttal, noting that the benchmark track does not require novel algorithms, and provided detailed responses, including expanded explanations of research directions, metric clarifications, and documentation updates.

Following the rebuttal, all reviewers maintained or raised their scores.  Overall, there is a clear consensus to accept the paper. I also see the code library as a unifying, helpful resource for the community.